# Shopping MMLU: A Massive Multi-Task Online Shopping Benchmark for Large Language Models

**Yilun Jin**[1,2], **Zheng Li**[1], **Chenwei Zhang**[1], **Tianyu Cao**[1], **Yifan Gao**[1], **Pratik Jayarao**[1]
**Mao Li**[1], **Xin Liu**[1], **Ritesh Sarkhel**[1], **Xianfeng Tang**[1], **Haodong Wang**[1], **Zhengyang Wang**[1]
**Wenju Xu**[1], **Jingfeng Yang**[1], **Qingyu Yin**[1], **Xian Li**[1], **Priyanka Nigam**[1], **Yi Xu**[1]
**Kai Chen**[2], **Qiang Yang**[2], **Meng Jiang**[1,3], **Bing Yin**[1]*
[1] Amazon.com       [2] HKUST       [3] University of Notre Dame
yilun.jin@connect.ust.hk, {amzzhe,cwzhang,caoty,yifangao,psj}@amazon.com
{maoamz,xliucr,rssarkhe,wanghaod,zhengywa}@amazon.com, tangxianfeng@outlook.com
{xuwenju,jingfe,qingyy,xianlee,nigamp,yxaamzn}@amazon.com
{kaichen,qyang}@cse.ust.hk, mjiang2@nd.edu, alexbyin@amazon.com

## Abstract

Online shopping is a complex multi-task, few-shot learning problem with a wide and evolving range of entities, relations, and tasks. However, existing models and benchmarks are commonly tailored to specific tasks, falling short of capturing the full complexity of online shopping. Large Language Models (LLMs), with their multi-task and few-shot learning abilities, have the potential to profoundly transform online shopping by alleviating task-specific engineering efforts and by providing users with interactive conversations. Despite the potential, LLMs face unique challenges in online shopping, such as domain-specific concepts, implicit knowledge, and heterogeneous user behaviors. Motivated by the potential and challenges, we propose Shopping MMLU, a diverse multi-task online shopping benchmark derived from real-world Amazon data. Shopping MMLU consists of 57 tasks covering 4 major shopping skills: concept understanding, knowledge reasoning, user behavior alignment, and multi-linguality, and can thus comprehensively evaluate the abilities of LLMs as general shop assistants. With Shopping MMLU, we benchmark over 20 existing LLMs and uncover valuable insights about practices and prospects of building versatile LLM-based shop assistants. Shopping MMLU can be publicly accessed at `https://github.com/KL4805/ShoppingMMLU`. In addition, with Shopping MMLU, we host a competition in KDD Cup 2024 [2] with over 500 participating teams. The winning solutions and the associated workshop can be accessed at our website `https://amazon-kddcup24.github.io/`.

## 1 Introduction

Machine learning (ML) has been applied to various user-oriented online services, such as online communities, streaming services, etc, with online shopping being among the most successful ones. In recent years, ML methods are applied to various online shopping tasks, such as user queries [25, 19, 15], sessions [45, 24], reviews [28, 27], product attributes [53, 38], etc. To facilitate the development of ML methods, many benchmarks are designed [16, 33] to lower the barrier for researchers and engineers to develop and evaluate novel solutions to real-world online shopping tasks.

---

*Work done partially during Yilun's internship at Amazon. Authors 4-15 ordered alphabetically. Correspondence: Yilun Jin (yilun.jin@connect.ust.hk), Zheng Li (amzzhe@amazon.com)

[2] `https://aicrowd.com/challenges/amazon-kdd-cup-2024-multi-task-online-shopping-challenge-for-llms`.

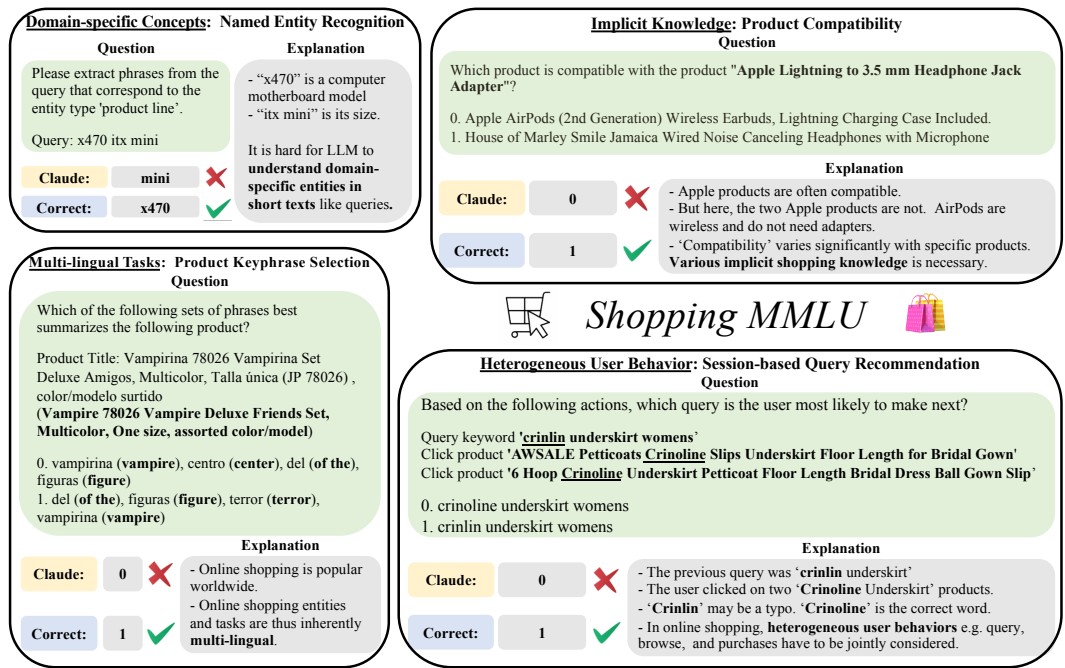

Figure 1: Distinctive characteristics of online shopping with real-world examples.

Online shopping is complex with numerous entities, relations, and tasks. For example, products are associated with *attributes, attribute values*, and *product categories*. Users interact with products with various behaviors such as *queries, clicks*, and *purchases*. Therefore, online shopping creates *multi-task learning* problems involving a joint understanding and modeling of these entities. Moreover, the entities and tasks are not fixed but expand over time with new users and services, such as the expansion of Amazon from shopping to streaming services, creating *few-shot* learning problems in the process. However, the multi-task, few-shot learning nature of online shopping is not sufficiently captured by existing works and benchmarks which mainly design task-specific models and datasets.

Large language models (LLMs) emerge as promising solutions to the multi-task, few-shot learning problem of online shopping. Recent works like GPT-3, T5, and FLAN [32, 5, 40] have shown that a single LLM can perform various text-related tasks with state-of-the-art performances, and can also generalize to unseen tasks with few-shot examples or even task descriptions only. These results motivate us to explore LLM-based solutions for online shopping with two advantages. First, we train a single LLM for all tasks instead of multiple task-specific models, which mitigates task-specific engineering efforts. Second, the trained LLM can seamlessly adapt to emerging tasks with only few-shot examples, lowering the costs for collecting large-scale data for model re-training. Moreover, LLM-based shop assistants can also improve user experiences by giving real-time interactive feedback to customer questions.

Despite the promising capabilities, LLM solutions for online shopping face specific challenges. We highlight the unique characteristics of tasks in online shopping with examples in Figure 1.

**Domain-specific Short Texts.** Texts in online shopping contain domain-specific entities, such as brands, models, etc., which may be challenging for general LLMs, especially without specific context.

**Implicit Knowledge.** Complex implicit knowledge and reasoning is required in online shopping to understand whether two products are compatible, or whether two brands produce similar items. Thus, it is challenging for LLMs to understand and adequately use the knowledge to perform reasoning.

**User Behaviors.** Aside from texts, implicit user behaviors exist (e.g. purchase, view, query-then-click, etc.) in online shopping. While implicit user behaviors are vital in understanding user intentions, general LLMs may not understand them as they rarely exist in pre-training data.

**Multi-lingual Tasks.** Online shopping spans a large number of countries, creating contents and tasks in multiple languages, which are challenging for LLMs trained with mostly English.

Table 1: Comparison between Shopping MMLU and related online shopping datasets. "Partially" means that the skill is covered with a limited number of tasks.

| Dataset | Unified Text-Gen Formulation | # Tasks | Concept Understanding | Knowledge Reasoning | User Behavior | # Languages |
|---|---|---|---|---|---|---|
| MAVE [48] | No | 1 | Partially | No | No | 1 |
| Amazon-M2 [16] | No | 3 | No | No | Partially | 6 |
| Amazon ESCI [33] | No | 3 | No | No | Partially | 3 |
| EComInstruct-Test (EcomGPT) [20] | Yes | 12 | Yes | No | No | 2 |
| ECInstruct (eCeLLM) [30] | Yes | 10 | Partially | No | Yes | 1 |
| **Shopping MMLU** | **Yes** | **57** | **Yes** | **Yes** | **Yes** | **6** |

Motivated by the above potentials and challenges, we propose Shopping MMLU, a diverse multi-task online shopping benchmark for LLMs. Shopping MMLU consists of 57 tasks and 20,799 questions curated with real-world Amazon data and covers an extensive range of shopping entities like products, categories, attributes, queries, reviews, sessions, etc. We re-formulate all tasks in Shopping MMLU as text-to-text generation to accommodate LLM-based solutions. Furthermore, to enable fine-grained analysis of model capabilities, we split Shopping MMLU into 4 shopping skills corresponding to the characteristics shown in Figure 1: *shopping concept understanding*, *shopping knowledge reasoning*, *user behavior alignment*, and *multi-lingual abilities*. We benchmark over 20 LLMs on Shopping MMLU to explore the potential of building LLM-based online shop assistants. Our experimental results uncover valuable insights for domain-specific LLMs in online shopping, such as task-wise correlations, transferability of general knowledge, effects of instruction fine-tuning, and in-context learning.

We believe that Shopping MMLU can inspire and facilitate the transition from task-specific efforts to versatile LLM-based methods in online shopping. Moreover, as the characteristics of online shopping (Figure 1) exist in other user-oriented services as well, we also expect that the insights uncovered by Shopping MMLU would benefit efforts to build domain-specific LLMs in a wider range of fields.

## 2   Related Work

**Online Shopping Datasets**   We summarize related online shopping datasets in Table 1. Previously, online shopping datasets often focus on one or several closely related tasks, e.g. MAVE [48] for attribute value extraction, Amazon-M2 [16] for session-based recommendation, Amazon-ESCI [33] for query-product matching, etc. Consequently, they fail to reflect the multi-task nature of online shopping as their coverage of tasks and skills is limited.

More recently, multi-task online shopping datasets are curated to build versatile LLM-based shop assistants, such as EComInstruct for EcomGPT [20] and ECInstruct for eCeLLM [30]. Both EComInstruct and ECInstruct reformulate online shopping tasks into text-to-text generation and fine-tune a single LLM to perform all tasks. However, despite being multi-task datasets, EComInstruct solely focuses on shopping concept understanding (12 out of 12 tasks), while ECInstruct primarily tackles user behavior alignment (8 out of 10 tasks). Therefore, their coverage of skills in online shopping is still limited compared to Shopping MMLU, especially in reasoning and multi-lingual abilities.

**LLMs for Online Shopping**   Shopping websites house various texts such as product titles, descriptions, reviews, ads, etc., motivating an extensive study of LLM solutions to online shopping tasks, such as recommendation [42, 18], ranking [12], named entity recognition [39], etc. However, these methods are limited to specific tasks without fully exploring the multi-task nature of LLMs.

More recent works leverage instruction fine-tuning (IFT) to adapt general domain LLMs to online shopping, such as EcomGPT and eCeLLM. However, as shown in Table 1, the capabilities of EcomGPT and eCeLLM may be limited as their training data cover a limited range of shopping tasks and skills.

**Web Agent Benchmarks for Online Shopping**   LLMs bring about exciting prospects in developing agents that can perform sequential decision making and task execution following text instructions. As online shopping websites are diverse, realistic, and interactive environments, many benchmarks are developed upon them, such as WebShop [49] and WebArena [54] where agents are required

to perform shopping tasks such as purchasing products and summarizing reviews. We believe that Shopping MMLU is complementary to these agent benchmarks, as agents should first gain sufficient knowledge of online shopping before executing composite decision making tasks.

# 3 Dataset and Task Description

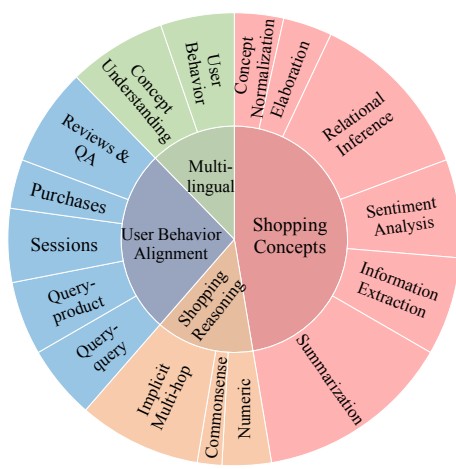

Figure 2: A brief taxonomy of Shopping MMLU including all skills and sub-skills.

In this section, we present the overall design of Shopping MMLU, featuring 57 tasks across 4 key skills based on real-world Amazon data. We present the raw data sources used, the task taxonomy, task designs, and evaluation metrics. Finally, we describe our efforts to improve the quality of Shopping MMLU.

## 3.1 Raw Data Sources

Shopping MMLU is curated primarily with real-world, internal or public [11, 8, 9, 16, 33] Amazon data, such as product catalogs, reviews, browse sessions, queries, etc. We remove all IDs (e.g. userID, sessionID, etc.) to ensure anonymity. We also use Claude 2 [1] to synthesize data for some tasks that do not involve concrete product or user data. Details of raw data sources are given in Appendix A.2.

## 3.2 Online Shopping Tasks

In this section, we introduce the task designs of Shopping MMLU, including the taxonomy, task types, and evaluation metrics.

### 3.2.1 Task Taxonomy

Shopping MMLU consists of 57 tasks across 4 shopping skills corresponding to Figure 1. Moreover, we divide each skill into sub-skills to enable more fine-grained evaluation. A simplified taxonomy is shown in Figure 2, with the full taxonomy in Figure 9 in the Appendix. We introduce each skill and their sub-skills as follows, and leave more details in Appendix A.3.

**Shopping Concept Understanding** ("Concept" for short). Online shopping concepts such as brands and product models are domain-specific and not often seen in pre-training. Moreover, they often appear in short texts (e.g. queries, attribute-value pairs) and thus no sufficient contexts are given to help understand them. Hence, failing to understand these concepts compromises the performance of LLMs on downstream tasks. We include the following sub-skills in this skill: *concept normalization*, *elaboration*, *relational inference*, *sentiment analysis*, *information extraction*, and *summarization*.

**Shopping Knowledge Reasoning** ("Reasoning" for short). This skill focuses on understanding and applying various implicit knowledge to perform reasoning over products and their attributes. For example, calculations such as the total volume of a product pack require numeric reasoning, and finding compatible products requires multi-hop reasoning among various products over a product knowledge graph. Based on the specific type of reasoning required, we split this skill into three sub-skills, *numeric*, *commonsense*, and *multi-hop* reasoning.

**User Behavior Alignment** ("Behavior" for short). Accurately modeling user behaviors is a crucial skill in online shopping. A large variety of user behaviors exist in online shopping, including queries, clicks, add-to-carts, purchases, etc. Moreover, these behaviors are generally implicit and not expressed in text. Consequently, LLMs trained with general texts encounter challenges in aligning with the heterogeneous and implicit user behaviors as they rarely observe such inputs during pre-training. We further design the following sub-skills to reflect such heterogeneous behaviors: *query-query relation*, *query-product relation*, *sessions*, *purchases*, and *reviews & QAs*.

**Multi-lingual Abilities** ("Multi-lingual" for short). Multi-lingual models are desired in online shopping as they can be deployed in multiple marketplaces without re-training. Therefore, we design the skill of multi-lingual online shopping, consisting of *multi-lingual concept understanding* and *multi-lingual user behavior alignment*.

### 3.2.2 Task Types

We include 5 types of tasks in Shopping MMLU for a comprehensive evaluation of shopping skills, including *multiple choice, retrieval, ranking, named entity recognition*, and *generation*. Due to different format requirements, each type of task requires specific prompts such that the evaluated LLMs follow the instructions and generate valid answers, which we show in Appendix A.5.

**Evaluation Metrics** We use *accuracy* for multiple choice tasks, *hit rate@3* for retrieval tasks, *normalized discounted cumulative gain (NDCG)* for ranking tasks, and *micro F1* for named entity recognition tasks. For generation tasks, we apply *ROUGE-L* scores for extraction tasks (i.e. the answer is a sub-string of the input), *BLEU* scores for translation tasks, and *sentence transformer similarity* [34] for other generation tasks. Details of the metrics are introduced in Appendix A.4. We take an average of all task-wise metrics (i.e. macro average) as the score of a skill.

### 3.3 Data Quality Control

Datasets of online shopping are either defined by human behaviors or are human-labeled, and thus may contain noise or errors. To address the issue, we manually inspect all data samples to ensure the validity of the questions. We also remove potentially offensive contents and all links to images and videos in product descriptions and reviews. Details of data filtering are described in Appendix A.6.

## 4 Experiments and Analyses

In this section, we present our experimental setup, results, and analyses based on Shopping MMLU. Our experiments uncover the following insights:

- Proprietary LLMs remain the state-of-the-arts on Shopping MMLU, with Claude-3 Sonnet performing the best overall. However, strong open-source LLMs have caught up with proprietary ones like ChatGPT.
- Tasks and skills in Shopping MMLU, and hence online shopping share much knowledge in common, as indicated by the highly positive correlations between pairwise tasks and skills in Shopping MMLU.
- General knowledge transfers well to the specific domain of online shopping. Strong models on general LLM benchmarks remain strong on Shopping MMLU.
- IFT improves the performance on Shopping MMLU in most cases. However, general domain IFT may lead to overfitting and hence compromise the contained knowledge in strong base models, while domain-specific IFT works only on strong base models and observed tasks and skills.
- Few-shot learning remains challenging on Shopping MMLU. In-context examples lead to worse performances for many models and tasks.

### 4.1 Experimental Setup

We apply zero-shot evaluation for Shopping MMLU for three main reasons. First, zero-shot evaluation resembles the real-world scenario where customers directly enter their questions without creating few-shot examples. Second, zero-shot evaluation rules out variances brought by different few-shot examples. Finally, all evaluated models achieve non-trivial results under zero-shot evaluation on Shopping MMLU. All models are tested with the same prompts.

### 4.2 Evaluated Models

We evaluate LLMs with various sizes and training methods to uncover insights about how to build domain-specific LLMs. Details of model access is given in Appendix B.1. Evaluated models include:

Table 2: Overall scores (%) on Shopping MMLU across all evaluated models. The best performances in LLMs with similar number of parameters are shown in **bold**.

| Model Type | # Params. | Model | Shopping Concept Understanding | Shopping Knowledge Reasoning | User Behavior Alignment | Multi-lingual Abilities |
|---|---|---|---|---|---|---|
| Proprietary | N/A | Claude-3 Sonnet | **80.75** | **71.63** | **70.17** | **67.76** |
| | | Claude-2 | 75.46 | 65.50 | 63.53 | 65.24 |
| | | ChatGPT | 75.63 | 64.97 | 59.79 | 60.81 |
| Open-Source | 70B | LLaMA3-70B-Instruct | **75.24** | **69.29** | **67.67** | 62.00 |
| | | QWen1.5-72B | 71.67 | 68.92 | 64.12 | **64.84** |
| | | LLaMA3-70B | 69.59 | 63.56 | 55.77 | 58.95 |
| | | LLaMA2-70B-chat | 61.84 | 40.73 | 44.20 | 47.04 |
| | | LLaMA2-70B | 61.05 | 55.87 | 43.24 | 47.85 |
| | | Mixtral-8x7b | 59.43 | 54.32 | 55.31 | 44.69 |
| | 14B | QWen1.5-14B | **67.22** | **60.92** | **54.92** | 55.21 |
| | | eCeLLM-L | 61.54 | 54.84 | 54.55 | **59.64** |
| | | Vicuna-13B | 59.64 | 52.63 | 49.81 | 49.64 |
| | | LLaMA2-13B-chat | 51.79 | 45.01 | 39.95 | 42.99 |
| | | LLaMA2-13B | 45.86 | 39.47 | 39.43 | 44.23 |
| | 7B | LLaMA3-8B-Instruct | **65.26** | **56.84** | **54.88** | 55.37 |
| | | LLaMA3-8B | 58.02 | 49.74 | 44.16 | 51.03 |
| | | QWen1.5-7B | 58.89 | 52.34 | 49.81 | 50.14 |
| | | eCeLLM-M | 63.29 | 48.94 | 53.78 | **56.08** |
| | | Zephyr | 61.65 | 52.57 | 44.73 | 45.35 |
| | | Mistral-7B-instruct | 62.03 | 46.36 | 42.21 | 43.32 |
| | | Mistral-7B | 55.82 | 46.69 | 46.27 | 41.47 |
| | | Vicuna-7B | 53.46 | 45.06 | 41.11 | 43.82 |
| | | LLaMA2-7B-chat | 51.67 | 43.48 | 41.42 | 40.43 |
| | | LLaMA2-7B | 38.22 | 32.81 | 32.56 | 27.71 |
| | <5B | QWen1.5-4B | **57.21** | **52.56** | **42.74** | **49.78** |
| | | Phi-2 | 49.34 | 42.83 | 36.38 | 32.91 |
| | | eCeLLM-S | 49.40 | 39.06 | 36.33 | 32.79 |

**Proprietary Models**   We evaluate ChatGPT [5], Claude-2 [1], and Claude-3 Sonnet [2], which are state-of-the-art LLMs trained with general domain data and provide insights on how well LLMs can solve domain-specific online shopping problems with general knowledge only.

**Open-Source General Models**   Open-source LLMs can be categorized as *base* and *chat models*. Base models refer to LLMs that are only pre-trained with next-token prediction without any moderation techniques, while chat models often undergo IFT such that they follow the input instructions. We include both base and chat models to see how the instruction following abilities of chat models transfer from the general domain to the specific domain of online shopping. Specifically, we consider **LLaMA2** (7/13/70B, base and chat) [37], **LLaMA3** (8/70B, base and instruct) [26], **Mistral** (7/8x7B, base and instruct) [14], **QWen1.5** (4/7/14/72B) [3], and **Phi-2** [13] models.

**Domain-specific Models**   We evaluate eCeLLM-S, M, and L models that are fine-tuned with domain-specific online shopping IFT data (ECInstruct [30]) over Phi-2, Mistral-7B, and LLaMA2-13B, respectively, to see how domain-specific IFT helps improve model performances on Shopping MMLU.

## 4.3   Overall Performance

We show the scores of all evaluated models on each skill of Shopping MMLU in Table 2. Due to space limitations, we omit detailed task-wise scores. We draw the following insights from Table 2.

First, **proprietary LLMs remain the state-of-the-art, while open-source LLMs are catching up**. Claude-3 Sonnet performs the best across all models, followed by Claude-2 and ChatGPT. Overall, these proprietary LLMs remain the strongest even in the specific domain of online shopping. We also observe that LLaMA3-70B-Instruct and QWen1.5-72B perform on par with ChatGPT and Claude-2, demonstrating the potential of building powerful LLM shop assistants with public resources.

Second, **Shopping MMLU is a challenging benchmark**. While eCeLLMs outperform GPT-4 on their dataset ECInstruct [30], they are still far behind ChatGPT on Shopping MMLU, showing that Shopping MMLU is a more complex and challenging benchmark for online shopping than ECInstruct.

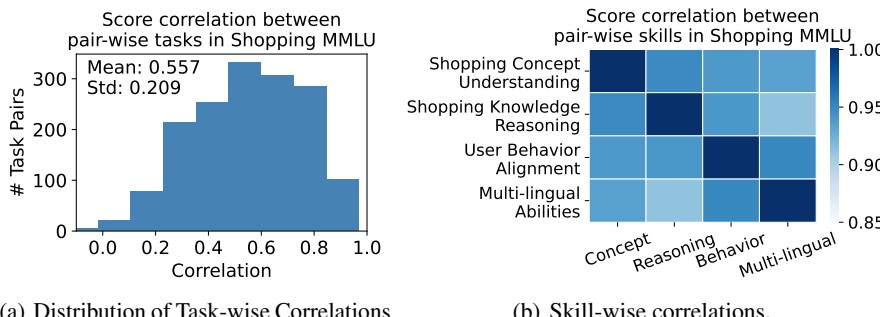

(a) Distribution of Task-wise Correlations

(b) Skill-wise correlations.

Figure 3: Task and skill-wise score correlations of Shopping MMLU.

Finally, **domain-specific models are not always strong**. While eCeLLMs perform better on Shopping MMLU than their base models (eCeLLM-M/Mistral-7B, eCeLLM-L/LLaMA2-13B), they are not always strong compared to LLMs with similar numbers of parameters. For example, among ∼13B LLMs, eCeLLM-L generally performs worse than QWen1.5-14B; among ∼7B LLMs, eCeLLM-M generally performs worse than LLaMA3-8B-Instruct. These facts indicate that LLMs with proper training in the general domain already excel in online shopping without domain-specific tuning.

### 4.4 How 'Multi-task' is Online Shopping?

According to [52], the key of multi-task learning is to leverage *useful information in multiple tasks* to improve *the performances of all tasks*. Consequently, the more the shared knowledge, the more likely we can jointly improve all tasks in Shopping MMLU and build versatile LLM-based shop assistants. Thus, in this section, we analyze the extent to which knowledge is shared among tasks in Shopping MMLU by analyzing the score correlations between pairwise tasks and skills.

We first analyze the task-wise score correlations. Let $\mathbf{s}_i$ be the scores achieved by all evaluated LLMs on task $i$, the score correlation between tasks $i$ and $j$ is defined as $c_{ij} = \mathrm{PearsonCorr}(\mathbf{s}_i, \mathbf{s}_j)$. The distribution of $c_{ij}$ is shown in Figure 3(a). As shown, the scores of most task pairs (1589 out of 1596) are positively correlated. Moreover, with an average of 0.557 and a standard deviation of 0.209, the score correlations are significantly positive, indicating a notable amount of shared knowledge among tasks in Shopping MMLU. We analyze task pairs with negative correlations in Appendix B.3.

We similarly compute the score correlations between pairwise skills and plot them in Figure 3(b). As shown, all skills are positively correlated with each other with correlations of at least 0.9. The observation further underscores the multi-task nature of Shopping MMLU and the potential of jointly improving online shopping skills as a whole with unified solutions.

### 4.5 How to Build LLM-based Shop Assistants?

In this section, we analyze various LLM moderation techniques, including model scaling, IFT, and in-context learning, to see whether and how they are helpful in improving the performances of LLMs on the specific domain of online shopping.

#### 4.5.1 General Knowledge Transfers Well to Online Shopping

The field of LLMs advances at a rapid pace, yielding models with increasingly powerful capabilities. Therefore, we analyze whether the specific domain of online shopping benefits from the advancing LLMs and their increasing general knowledge and abilities. We calculate the score correlations between each skill in Shopping MMLU and the Open LLM Leaderboard [4], consisting of MMLU, GSM8K, Winogrande, HellaSwag, TruthfulQA, and ARC [10, 7, 23, 50, 36, 6]. The correlations are shown in Figure 4(a), where all skills show strongly positive correlations with the Open LLM Leaderboard scores. The high correlations indicate that general knowledge transfers well to the specific domain of online shopping, and that powerful LLM-based shop assistants should be established upon strong base models.

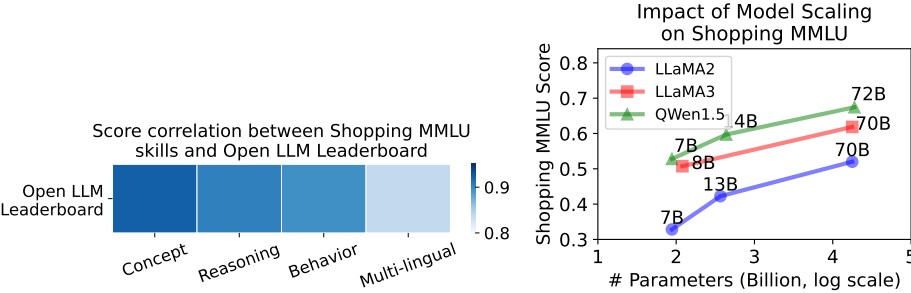

(a) Score correlations between Shopping MMLU skills and Open LLM Leaderboard [4]

(b) Effects of model scaling on Shopping MMLU

Figure 4: General knowledge transfers well to online shopping.

The smooth transfer from general knowledge to the domain of online shopping is also observed in the effects of model scaling, which are shown in Figure 4(b). We observe consistent improvements on Shopping MMLU as LLMs within each family (LLaMA2, LLaMA3, and QWen1.5) increase in size.

### 4.5.2 Effects of Instruction Fine-tuning

In this section, we analyze the effects of IFT [40] on Shopping MMLU. We analyze both general domain and domain-specific IFT to understand whether the instruction following ability transfers from the general domain to online shopping, and how domain-specific IFT achieves further improvement.

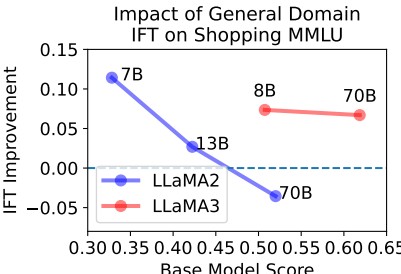

Figure 5: Relation between base model scores and improvements of IFT on Shopping MMLU.

**General domain IFT**   We analyze LLaMA2 and LLaMA3 models to study the impact of general domain IFT on Shopping MMLU. We plot the relation between scores of base models and the improvements brought by IFT (e.g. LLaMA3-8B-Instruct VS Base) in Figure 5. We only plot the average values across all 4 skills and leave details in Appendix B.4. We make the following observations.

First, **general domain IFT helps in most cases**. Among the 5 models tested, IFT leads to performance improvements on 4 of them, indicating that the instruction following ability brought by general domain IFT often transfers to the specific domain of online shopping. Second, **IFT data and recipe matters**. Comparing LLaMA2 and LLaMA3, we find that LLaMA3 models generally benefit more from IFT, which can be attributed to the better instruction data with 'careful curation' used to tune LLaMA3 [26]. Finally, **general domain IFT is less helpful on stronger base models.** Within each model family, IFT leads to less improvements on stronger base models. Notably, IFT leads to performance decline on LLaMA2-70B. We hypothesize that as base models gets stronger, they may overfit to the relatively small IFT dataset during IFT, resulting in the catastrophic forgetting of helpful knowledge.

**Domain-specific IFT**   As shown in Table 2, while eCeLLMs perform better than their base models with domain-specific IFT, they do not compare favorably against strong general domain LLMs (LLaMA3 and QWen1.5). Therefore, we analyze the reasons underlying the limited improvements and shed light on how domain-specific IFT data should be curated. We show the comparisons between eCeLLMs and their base models in Figure 6. We also include Zephyr and Vicuna-13B, which are tuned with general domain IFT over Mistral-7B and LLaMA2-13B, respectively. We make the following observations.

- **Domain-specific IFT only works on strong base models**. As shown in Figure 6(a), eCeLLM-S fails to improve over its base model Phi-2, while in Figure 6(b) and 6(c), both eCeLLM-M and

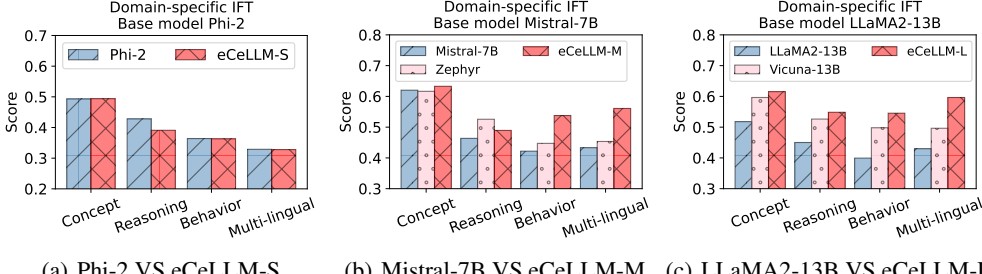

(a) Phi-2 VS eCeLLM-S    (b) Mistral-7B VS eCeLLM-M    (c) LLaMA2-13B VS eCeLLM-L

Figure 6: Comparison between domain-specific eCeLLMs and their base models on Shopping MMLU.

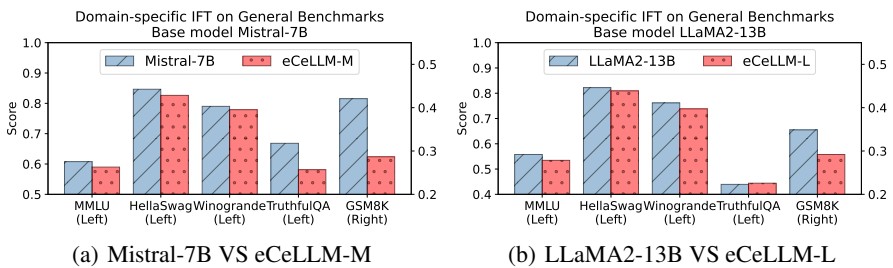

(a) Mistral-7B VS eCeLLM-M    (b) LLaMA2-13B VS eCeLLM-L

Figure 7: Scores of eCeLLM and their base models on general LLM benchmarks.

L outperform their base models. The observation indicates that domain-specific IFT works only on sufficiently strong base models, which echoes the phenomenon in general domain IFT [40].

- **Domain-specific IFT only works on observed tasks and skills**. As shown in Figure 6(b) and 6(c), eCeLLMs primarily improve over their counterparts on "Behavior" and "Multi-lingual" skills, which should be attributed to their IFT datasets. In ECInstruct used to tune eCeLLMs, 8 out of 10 tasks belong to the "Behavior" skill. Therefore, it is not surprising that eCeLLMs perform well on the "Behavior" skill. We also hypothesize that the knowledge transfer from English to other languages leads to the improvement on the "Multi-lingual" skill, as this skill consists heavily of multi-lingual user behavior alignment tasks. However, eCeLLMs achieve limited improvements on "Concept" and "Reasoning" skills, showing that domain-specific IFT only works on skills included in the IFT data and does not generalize well to unseen skills. Therefore, domain-specific IFT data should be curated with sufficient diversity and coverage.

We also test eCeLLM-M and L on 5 general LLM benchmarks, MMLU, HellaSwag, Winogrande, TruthfulQA, and GSM8K to analyze why domain-specific IFT fails to generalize to unseen skills. Results are shown in Figure 7, where eCeLLMs perform worse than their base models in most general LLM benchmarks. Thus, domain-specific IFT fails to improve or even compromises the model's general knowledge, which may explain their inability to generalize to unseen skills.

### 4.5.3 Effects of In-context Learning

LLMs are capable of learning from few-shot examples in prompts, known as *in-context learning*. As few-shot learning is common in online shopping, such as cold-start users, we analyze how well LLMs adapt to unseen tasks with few-shot examples and thus solve the few-shot learning problem. We select representative subsets of models and tasks for the analysis (details in Appendix B.5). For each selected task, we split the dataset into a training set of 20 samples, and the rest as test sets. We evaluate under 0-, 1-, and 5-shot settings and show results in Figure 8. For each setting, we randomly sample few-shot examples from the training set and show the mean score of 5 random seeds. We observe the following phenomena despite the mixed results.

First, **in-context learning is not generally helpful on Shopping MMLU.** We observe that in many cases, adding few-shot examples fails to improve model performances. Even worse, for some models

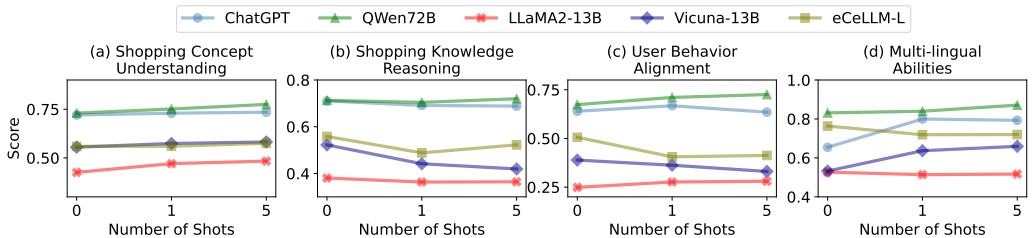

Figure 8: Results of in-context learning (0-, 1-, and 5-shot) on representative tasks in Shopping MMLU.

and skills, in-context examples lead to worse scores (e.g. ChatGPT, Vicuna-13B and eCeLLM-L in Figure 8(c)). The observation indicates that few-shot learning in online shopping remains challenging even with strong LLMs. Second, **in-context learning does not help reasoning tasks.** We observe from Figure 8(b) that in-context learning fails to improve the performance of any model on shopping knowledge reasoning tasks. We further explore this observation with chain-of-thought (CoT) prompting [41], whose results are shown in Appendix B.5.

## 5 Conclusion and Future Work

This paper presents Shopping MMLU, a multi-task online shopping benchmark for LLMs aiming to facilitate LLMs-based solutions to a unified, multi-task modeling of online shopping. Shopping MMLU features a wide range of online shopping skills, tasks, and entities, and thus is suitable for researchers and practitioners to comprehensively evaluate their solutions of domain-specific LLM online shop assistants. With Shopping MMLU, we perform extensive experiments on over 20 LLMs, whose results uncover valuable insights on building domain-specific LLMs for online shopping, such as task- and skill-wise relations, general knowledge, instruction fine-tuning, and in-context learning.

Shopping MMLU triggers a series of future work. In Appendix C we show that state-of-the-art proprietary LLMs still lags behind task-specific methods on some tasks of Shopping MMLU, motivating advanced training recipes and data for LLMs in online shopping. We also discuss broader impacts and limitations in Appendix C.

## Acknowledgments and Disclosure of Funding

We thank AWS and amazon science for their support. This work is partially supported by Hong Kong RGC TRS T41-603/20-R and the Turing AI Computing Cloud at HKUST [46].

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

# A    More Dataset and Task Details

## A.1    License

Following the licenses of similar datasets [16, 8], Shopping MMLU can be freely used under the license of Apache 2.0.

## A.2    Data Sources

We summarize the data sources we use in this Section.

### A.2.1    Public Data from Amazon

Some tasks of Shopping MMLU are created with public data from Amazon, including:

- Amazon-M2 [16], including browse sessions and product metadata from 6 marketplaces (United Kingdom (UK), Spain (ES), Germany (DE), Japan (JP), France (FR), and Italy (IT)).
- Amazon-ESCI [33], including user queries, related products, as well the relevance between the query and the products. The data come in 3 languages, English, Japanese, and Spanish.
- Amazon Product Keyphrases [8], including product metadata (title, descriptions) as well as product keyphrases derived from users' queries. The data come in 5 languages, English, German, Spanish, French, and Italian.
- Amazon Reviews [11], including product metadata, user reviews to products, as well as various tags such as the number of upvotes received by each review, product-product relations (also-buy, also-view), etc.
- Amazon QA [9], including user-generated questions and answers about products, as well as product reviews that may be related to the question. The goal of the dataset is to automatically generate answers to user questions based on contexts in the reviews.

### A.2.2    Internal Data

Shopping MMLU is primarily constructed with internal data from Amazon. They can be roughly categorized into four classes.

- **Catalog Data**, which contains the ontology of Amazon to organize products. Catalog data contains the hierarchy, meanings, and relationships (e.g. applicability, complementarity) of product categories, attributes, attribute values, etc.
- **Product Data**, which contains product metadata such as attributes and values, product categories, titles, descriptions, etc.
- **Review Data**, which contains user reviews to products along with fine-grained labels like aspects, sentiments, and keyphrases.
- **Browse Data**, which contains user behaviors such as sessions, queries, as well as datasets derived from user behaviors (e.g. query-product category relations, query-attribute relations, etc.).

We remove all identifiers (e.g. sessionID, userID, productID, reviewID) for all internal data to maintain anonymity. We have obtained approval from the Amazon legal team to publish the data.

### A.2.3    Synthetic Data

We use Claude-2 to synthesize data for tasks that do not involve specific products, users, etc., including:

- **Unit conversion**. We sample a set of units that are used in shopping and ask Claude-2 to generate unit conversion questions within these units.
- **Shopping Commonsense**. We use Claude-2 to sample questions from ATOMIC10X [43] that are related to shopping and products.

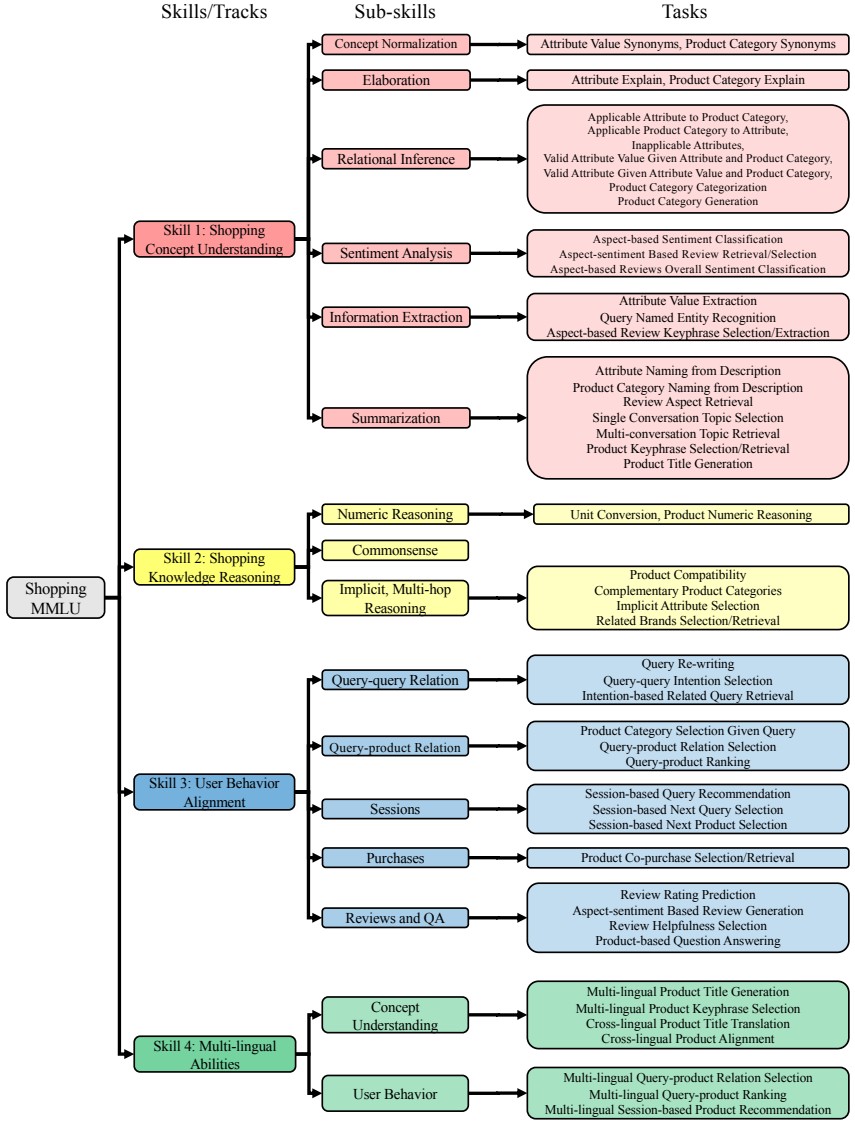

Figure 9: Full taxonomy of Shopping MMLU including skills, sub-skills, and tasks.

- **Conversation Topics**. We generate synthetic conversations between a user and a shop assistant with Claude-2. The conversations begin with a seed (i.e. a shopping intention) and goes on iteratively between two Claude-2 models.

### A.3 Full Task Taxonomy

We provide detailed introduction of each sub-skill as follows.

**Shopping Concept Understanding**  . We include the following sub-skills in this skill.

1. **Concept Normalization**, which measures the ability to unify terms with the same meanings. For example, 'USB3.0', 'USB3.1Gen 1', and 'USB3.2 Gen1' refer to the same USB standards.

2. **Elaboration**, which tests the model's ability to explain products in plain and understandable languages to facilitate customer shopping decisions.

3. **Extraction** and **Summarization**, which focuses on the model's ability to extract specific details or provide concise and informative summaries from long product descriptions.

4. **Relational Inference**, which focuses on the compatibility and interactions between concepts (e.g. product category and attribute, attribute and attribute value, etc.).

5. **Sentiment Analysis**, which requires the model to extract fine-grained aspects and sentiments from customer reviews, and thus recommend users with high-quality products.

**Shopping Knowledge Reasoning**  ("Reasoning" for short). Based on the type of reasoning required, this skill is divided into:

1. **Numeric Reasoning**, where the LLM extracts necessary numeric information from product metadata and perform calculations to derive results.

2. **Commonsense Reasoning**, which tests the model's ability to infer and reason over commonsense knowledge of daily products (e.g. intended usage and purpose).

3. **Multi-hop Reasoning**, which requires the model to draw connections across multiple entities and relations in online shopping to get the answer (e.g. similarity, compatibility, complementarity, etc.).

**User Behavior Alignment**  ("Behavior" for short). Accurately modeling user behaviors is a crucial skill in online shopping. Various kinds of user behaviors exist in online shopping, including queries, clicks, add-to-carts, purchases, etc. Moreover, these behaviors are generally implicit and not expressed in languages. Consequently, LLMs trained with general texts encounter challenges in aligning with the heterogeneous and implicit user behaviors as they rarely observe such inputs during pre-training. We further design the following sub-skills, each of which focuses on a specific type of user behavior.

1. **Queries**. Most online shopping experiences starts with a user query that reflect the user's initial intentions. Afterwards, the user may either initiate other related queries, or browse products that meet his intentions. We thus include two sub-skills corresponding to both scenarios, **query-query relation** and **query-product relation**.

2. **Sessions**, which evaluates how well the model understands a user's short-term shopping interests and recommend the user with the next possible product or query.

3. **Purchases**, which focuses the model's ability to help users directly make purchase decisions without the arduous process of searching and browsing.

4. **Reviews and QAs**, which requires the model to provide helpful feedbacks to various user-generated contents on an online shopping platform, such as answering product-related questions, and casting votes to informative reviews.

**Multi-lingual Abilities**  ("Multi-lingual" for short). We include sub-skills of **multi-lingual shopping concept understanding**, and **multi-lingual user behavior alignment** in this skill, which are multi-lingual versions of the corresponding skills, respectively.

The full taxonomy containing skills, sub-skills and tasks are shown in Figure 9. Detailed task descriptions are shown in Table 3, 4, 5, and 6 for each skill.

Table 3: Summary of tasks and datasets in the skill "Shopping Concept Understanding".

| Sub-skill | Task Name | Description | Task Type | Metric | # Samples |
|---|---|---|---|---|---|
| Concept Normalization | Product Category Synonyms Selection | Select a synonymous phrase to the given product category. | Multiple Choice | Accuracy | 234 |
| | Attribute Value Synonyms Selection | Select a synonymous phrase to the given attribute value. | Multiple Choice | Accuracy | 290 |
| Elaboration | Attribute Explain | Explain a given attribute. | Generation | Sentence transformer similarity | 300 |
| | Product Category Explain | Explain a given product category. | Generation | Sentence transformer similarity | 184 |
| Relational Inference | Applicable Attribute Selection Given Product Category | Given a product category, select an attribute that is applicable to the product category. | Multiple Choice | Accuracy | 884 |
| | Applicable Product Category Selection Given Attribute | Given an attribute, select a product category so that the attribute applies to it. | Multiple Choice | Accuracy | 843 |
| | Inapplicable Attributes | Given a product and a question on an attribute that does not apply to it, correctly answer 'not apply'. | Multiple Choice | Accuracy | 206 |
| | Valid Attribute Value Selection Given Attribute and Product Category | Given a product category and an attribute, select an appropriate attribute value. | Multiple Choice | Accuracy | 1152 |
| | Valid Attribute Selection Given Attribute Value and Product Category | Given a product category and an attribute value, select an appropriate attribute with the value. | Multiple Choice | Accuracy | 1152 |
| | Product Category Classification | Select the appropriate product category given a product's metadata. | Multiple Choice | Accuracy | 820 |
| | Product Category Generation | Generate the appropriate product category given a product's metadata. | Generation | Sentence transformer similarity | 525 |
| Sentiment Analysis | Aspect-based Sentiment Classification | Classify the sentiment of a review with respect to an aspect. | Multiple Choice | Accuracy | 395 |
| | Aspect-sentiment-based Review Retrieval | Retrieve reviews from a candidate list according to an aspect and sentiment. | Retrieval (3-in-15) | Hit rate @ 3 | 171 |
| | Aspect-sentiment-based Review Selection | A multiple choice task with similar requirements as above. | Multiple Choice | Accuracy | 346 |
| | Aspect-based Review Overall Sentiment Classification | Select the overall sentiment of a list of 20 reviews on a given aspect. | Multiple Choice | Accuracy | 424 |
| Information Extraction | Attribute Value Extraction | Extract value of a certain attribute given product metadata. | Multiple Choice | Accuracy | 338 |
| | Query Named-entity Recognition | Extract named-entities of a certain type from a user query. | Named entity recognition | Micro-F1 | 361 |
| | Aspect-based Review Keyphrase Extraction | Extract a keyphrase from a review that corresponds to the aspect. | (Extractive) Generation | ROUGE-L | 200 |
| | Aspect-based Review Keyphrase Selection | A multiple choice task with similar requirements as above. | Multiple Choice | Accuracy | 384 |
| Summarization | Attribute Naming from Description | Generate the attribute name given the descriptions to it. | Generation | Sentence transformer similarity | 300 |
| | Product Category Naming from Description | Generate the product category name given the descriptions to it. | Generation | Sentence transformer similarity | 213 |
| | Review Aspect Retrieval | Retrieve all aspects mentioned by the review from a candidate list. | Retrieval | Hit rate @ 3 | 200 |
| | Single Conversation Topic Selection | Select the most appropriate topic for a shopping-related conversation. | Multiple Choice | Accuracy | 299 |
| | Multi-conversation Topic Retrieval | Retrieve the topics covered by 3 shopping-related conversations. | Retrieval | Hit rate @ 3 | 250 |
| | Product Keyphrase Selection | Select the set of keyphrases that best describes the given product. | Multiple Choice | Accuracy | 233 |
| | Product Keyphrase Retrieval | A retrieval task with similar requirements as above. | Retrieval | Hit rate @ 3 | 233 |
| | Product Title Generation | Generate an adequate title given product metadata. | Generation | Sentence transformer similarity | 193 |
| **Total** | | | | | **11129** |

Table 4: Summary of tasks and datasets in the skill "Shopping Knowledge Reasoning".

| Sub-skill | Task Name | Description | Task Type | Metric | # Samples |
|---|---|---|---|---|---|
| Numeric Reasoning | Unit Conversion | Convert a quantity from one unit to another. | Multiple Choice | Accuracy | 390 |
| | Product Numeric Reasoning | Given a product, answer a question with numerical reasoning (add or multiply). | Multiple Choice | Accuracy | 493 |
| Commonsense Reasoning | Commonsense | Product-based commonsense question answering. | Multiple Choice | Accuracy | 463 |
| Implicit Multi-hop Reasoning | Complementary Product Categories | Select a product category that complements the given product category | Multiple Choice | Accuracy | 546 |
| | Implicit Attribute Selection | Given a user query, select an attribute value that is not explicitly in the query. | Multiple Choice | Accuracy | 552 |
| | Product Compatibility | Select a compatible product with the given product. | Multiple Choice | Accuracy | 141 |
| | Related Brands Selection | Select a brand that is similar or related with the given brand. | Multiple Choice | Accuracy | 266 |
| | Related Brands Retrieval | A retrieval task with similar requirements as above. | Retrieval | Hit rate @ 3 | 2661 |
| **Total** | | | | | **3117** |

Table 5: Summary of tasks and datasets in the skill "User Behavior Alignment".

| Sub-skill | Task Name | Description | Task Type | Metric | # Samples |
|---|---|---|---|---|---|
| Query-query Relation | Query Re-writing | Re-write a given query according to a required aspect and a value. | Generation | Sentence transformer similarity | 439 |
| | Query-query Intention Selection | Given a user query and a follow-up query, select the shopping intention. | Multiple Choice | Accuracy | 600 |
| | Intention-based Related Query Retrieval | Given a user query and a shopping intention, retrieve related queries from a candidate list. | Retrieval | Hit rate @ 3 | 300 |
| Query-product Relation | Product Category Selection Given Query | Given a user query, select a product category that the user may purchase. | Multiple Choice | Accuracy | 249 |
| | Query-product Relation Selection | Given a query and a product, select the relation between them. | Multiple Choice | Accuracy | 280 |
| | Query-product Ranking | Given a query and a list of products, rank them according to their relevance to the query. | Ranking | NDCG | 150 |
| Sessions | Session-based Query Recommendation | Given a user session with queries and product browses, retrieve the next query the customer may make. | Retrieval | Hit rate @ 3 | 60 |
| | Session-based Next Query Selection | A multiple choice task with similar requirements as above. | Multiple Choice | Accuracy | 60 |
| | Session-based Next Product Selection | Given a user browse session, select the next product the user will view. | Multiple Choice | Accuracy | 120 |
| Purchase | Product Co-purchase Selection | Given a product, select another product that is often purchased with it | Multiple Choice | Accuracy | 375 |
| | Product Co-purchase Retrieval | A retrieval task with similar requirements as above. | Retrieval | Hit rate @ 3 | 250 |
| Reviews & QA | Review Rating Prediction | Given a piece of review text, predict the customer rating. | Multiple Choice (1-in-5) | Accuracy | 552 |
| | Aspect-sentiment-based Review Generation | Given a product and aspect-sentiment pairs, generate an adequate review. | Generation | Sentence transformer similarity | 190 |
| | Review Helpfulness Selection | Given four reviews to the same product, select the one with the most votes. | Multiple Choice | Accuracy | 217 |
| | Product-based Question Answering | Given a question and some review texts of a product, answer the question. | Generation | Sentence transformer similarity | 131 |
| **Total** | | | | | **3973** |

Table 6: Summary of tasks and datasets in the skill "Multi-lingual Abilities". DE=German, ES=Spanish, FR=French, IT=Italian, JP=Japanese.

| Sub-skill | Task Name | Description | Task Type | Metric | # Samples | Languages |
|---|---|---|---|---|---|---|
| Concept Understanding | Multi-lingual Product Title Generation | Generate a product title given multi-lingual product metadata. | Generation | Sentence transformer similarity | 284 | DE, ES, FR, IT |
| | Multi-lingual Product Keyphrase Selection | Select the set of multi-lingual keyphrases that best describe the product. | Multiple Choice | Accuracy | 400 | DE, ES, FR, IT |
| | Cross-lingual Product Title Translation | Translate a product title from English to another language. | Generation | BLEU score | 500 | DE, ES, FR, IT, JP |
| | Cross-lingual Product Alignment | Given a product metadata, select the product metadata in another language. | Multiple Choice | Accuracy | 300 | DE, ES, FR, IT, JP |
| User Behavior | Multi-lingual Query-product Relation Selection | Select the relation between a multi-lingual query and product. | Multiple Choice | Accuracy | 320 | ES, JP |
| | Multi-lingual Query-product Ranking | Rank a list of multi-lingual products according to their relevance to a query. | Ranking | NDCG | 200 | ES, JP |
| | Multi-lingual Session-based Next Product Selection | Given a user browse session, select the next product the user will view. | Multiple Choice | Accuracy | 375 | DE, ES, FR, IT, JP |
| **Total** | | | | | **2379** | |

## A.4 Metrics

In this section, we provide detailed descriptions of our metrics.

- **Multiple Choice**: We follow the HELM [21] style of evaluating multiple choice questions. Specifically, we let the model generate one token and compare it with the answer to calculate *accuracy*.

- **Retrieval**: We let the model generate three comma-separated numbers (e.g. "1, 2, 3"), split the generation with comma, and compare the retrieved list with the ground truth to calculate *hit rate@3*. We set the number of retrieved instances as 3 because all retrieval tasks in Shopping MMLU has fewer than 3 positive examples. Let $retr$ denote the retrieved set of instances, and $truth$ denote the ground truth, $hit\ rate@3$ is calculated as

$$hit\ rate@3 = \frac{|truth \cap retr|}{|truth|}. \tag{1}$$

- **Ranking**: Each ranking question is provided with a query and 5 candidate samples, and each candidate is assigned a relevance score to the query. The model is asked to generate a permutation from 1 to 5, separated with comma (e.g. 1, 2, 3, 4, 5), which we will split with comma and obtain the re-ranked list. Let $rank_i$ denote the $i$-th sample in the re-ranked list, $rel(\cdot)$ denote the relevance of the sample, NDCG is calculated as

$$\begin{aligned} \text{DCG} &= \sum_{i=1}^{5} \frac{rel(rank_i)}{\log_2(i+1)}, \\ \text{NDCG} &= \frac{\text{DCG}}{\text{iDCG}}, \end{aligned} \tag{2}$$

where iDCG (the ideal DCG) is defined as the DCG achieved when the samples are ranked in descending order. Thus, $\text{NDCG} \in (0, 1]$.

- **Named Entity Recognition**. We use Micro-F1 score to evaluate named entity recognition tasks. Specifically, we let $\text{TP}, \text{FP}, \text{FN}$ denote true positives, false positives (recognizing non-existing entities), and false negatives (failure to recognize entities), and calculate Micro F1 as,

$$\text{Precision} = \frac{\text{TP}}{\text{TP} + \text{FP}}, \text{Recall} = \frac{\text{TP}}{\text{TP} + \text{FN}}, \text{F1} = \frac{2 \cdot \text{Precision} \cdot \text{Recall}}{\text{Precision} + \text{Recall}}. \tag{3}$$

- **Generation**. For extractive generation, we adopt ROUGE-L scores (F1) [22]. For translation scores, we adopt BLEU-4 scores [29] based on the package `sacrebleu` [31]. For other unrestricted generation tasks, we adopt sentence transformers [34] to first transform the generated texts and the reference texts into embeddings $\mathbf{x}_{gen}, \mathbf{x}_{ref}$, and then compute the cosine similarity $\frac{\mathbf{x}_{gen}^T \mathbf{x}_{ref}}{\|\mathbf{x}_{gen}\|\|\mathbf{x}_{ref}\|}$ as the metric. Empirically, the cosine similarity is rarely negative, and we set the score to 0 if it happens. We are aware that there are other metrics

for evaluating text generation, such as BERT Score [51]. As BERT score correlates well with sentence transformer similarity (>0.85) but varies significantly less (almost all BERT scores are greater than 0.8), we adopt sentence transformer similarity.

Table 7: Sample prompt of multiple choice questions.

| Task: Product Type Synonym |
| --- |
| Which of the following products is designed for a different purpose than promoting healthy hair?
0. Hair care product
1. Hair product
2. Hair cair agent
3. Nail Polish
Answer: |
| Correct Answer: 3 |

Table 8: Sample prompt of retrieval questions.

| Task: Related Keywords Retrieval |
| --- |
| A user on an e-commerce platform has just made a query.
The user wants to make another query with a shopping intention
(narrowing, substitute, or complement).
You are given a list of 15 numbered queries.
Choose three queries that the user is most likely to make
according to the previous query and the intention.
You should output three numbers, separated by comma. Do not give explanations.
Previous Query: white cardigan for women
Intention: narrowing
Query List:
1. white camisoles for women
2. white jean jacket women
3. white button down shirt women
4. orange throw blanket
5. white shrug for women
6. white cardigan for women summer
7. white cardigan for women short sleeve
8. mattress cover full
9. black cardigan for women
10. tide free and gentle laundry detergent
11. green bag
12. platform crocs
13. cream cardigan for women
14. frog hat
15. white cardigan for women dressy
Output: |
| Correct Answer: 6, 7, 15 |

## A.5 Sample Prompts

In this section, we show sample prompts for multiple choice, retrieval, ranking, named entity recognition, and generation tasks in Table 7, 8, 9, 10, and 11, respectively. By default, we use number choices for multiple choice questions. However, as the numbers may be confused with decimal points, we use letter choices for tasks involving numeric reasoning (example shown in 12). All questions are evaluated with a prepended system prompt:

- "You are a helpful online shopping assistant. Please answer the following question about online shopping and follow the given instructions and examples. "

Table 9: Sample prompt of ranking questions.

| Task: Query Product Ranking |
| --- |
| You are an intelligent shopping assistant that can rank products based on their relevance to the query. |
| The following numbered list contains 5 products. |
| Please rank the products according to their relevance with |
| the query 'super radio 3 amfm high-performance super radio'. |
| Product List: |
| 1. Wireless Bluetooth Speaker 4.0 Speaker Stereo Strong Enhanced Bass FM Radio MP3 Player (Gray) |
| 2. Monster Rockin' Roller Charge Bluetooth Speaker |
| 3. Amazon Basics 16-Gauge Speaker Wire Cable, 100 Feet |
| 4. RCA RP7887 Super Radio 3 |
| 5. Amazon Basics 12 Pack D Cell All-Purpose Alkaline Batteries, 5-Year Shelf Life, Easy to Open Value Pack |
| You should output a permutation of 1 to 5. There should be a comma separating two numbers. |
| Each product and its number should appear only once in the output. |
| Only respond with the ranking results. Do not say any word or explanations. |
| Output: |
| Correct Answer: 4, 1, 3, 5, 2 |

Table 10: Sample prompt of named entity questions.

| Task: Query Named Entity Recognition |
| --- |
| You are a helpful online shop assistant and a linguist. |
| A customer on an online shopping platform has made the following query. |
| Please extract phrases from the query that correspond to the entity type 'brand'. |
| Please directly output the entity without repeating the entity type. |
| If there are multiple such entities, separate them with comma. Do not give explanations. |
| Query: sigma lens for canon |
| Output: |
| Correct Answer: sigma |

## A.6 Details of Data Filtering

We introduce our efforts to filter the raw data and curate Shopping MMLU as follows. Due to the varying nature of tasks in Shopping MMLU, many of these efforts are task-specific.

- **Product Category Generation**. We remove all products where its 'product category' exists in its title and metadata.

- **Aspect-based Sentiment Classification**. In many cases, the sentiment towards an aspect expressed in a review is mixed, i.e. there are both positive and negative mentions. To avoid ambiguity, we select 'positive' and 'negative' samples as reviews that have solely positive/negative mentions on an aspect.

- **Aspect-sentiment-based Review Retrieval**. Many reviews snippets are vague and can be associated with many aspects, such as 'works great', 'looks good', 'nice product'. We manually check the questions to filter out these vague review snippets.

- **Attribute Value Extraction**. The raw data includes attribute, attribute values, as well as product metadata. However, in many cases the attributes cannot be derived from the given metadata. We perform manual inspection to make sure that all attributes can be found in the given product information.

- **Aspect-based Review Keyphrase Extraction**. Many reviews mention an aspect more than once. To avoid ambiguity (and to reduce the task difficulty), we select reviews and aspects, such that the aspect is only mentioned once in the review.

- **Product Title Generation**. Many products do not have sufficiently long metadata (i.e. product description) to support generating an informative title. We manually remove these products from this task.

- **Product Numeric Reasoning**. Similar to 'Attribute Value Extraction', in many cases, the attributes cannot be derived from the given metadata. Moreover, the numeric attributes

Table 11: Sample prompt of generation questions.

| **Task: Product Title Generation** |
| --- |
| Please generate an adequate title for the product with the following descriptions. Product Descriptions: Quest Salted Caramel protein shakes are simply made with 11 ingredients. The end result is a delicious, naturally flavored shake that provides your body with 30g of protein, 3 grams of carbs, and 1 gram of sugar. Our non-GMO shakes are custom-made and mixed to perfection to ensure every sip is as delicious as your cravings. Each shake has 30g of protein, 3-4g carbs and 1g of sugar - and is naturally flavored and non-GMO Output: |
| **Sample Answer:** Quest Nutrition Ready to Drink Salted Caramel Protein Shake, High Protein, Low Carb, Gluten Free, Keto Friendly |

Table 12: Sample prompt of multiple choice questions with letter choices.

| **Task: Product Numeric Reasoning** |
| --- |
| The product 'MADHAVA Organic Light Agave, 46 oz. Bottle (Pack of 2) \| 100% Pure Organic Blue Agave Nectar \| Natural Sweetener, Sugar Alternative \| Vegan \| Organic \| Non GMO \| Liquid Sweetener' appears on e-commerce website. What is the total volume of the two bottles of agave nectar? (A) 25.4 fl oz (B) 202.8 fl oz (C) 26 fl oz (D) 92 fl oz Please answer the question with a single letter indicating your choice. Answer: |
| Correct Answer: D |

themselves are not accurate in some cases. We check for both types of noises and filter questions accordingly.

- **Implicit Attribute Selection**. The raw data for this task is derived from customer behaviors (i.e. common attributes of clicked products after a query), and thus is very noisy. We manually check for the validity of the implicit attributes.

- **Session-based Query/Product Recommendation**. We manually inspect all sessions and remove sessions with abrupt changes in shopping intentions. We empirically observe that all models perform better after the filtering.

- **Review Helpfulness Selection**. We remove reviews with images and videos as these additional information also contributes to 'helpfulness'. We also remove reviews that have an abnormal number of 'helpfulness' votes (e.g. the 'most helpful' review has >2000 votes, while the remaining ones have about 100 votes). We empirically observe consistently better performances after removing such reviews.

- **Product-based Question Answering**. This task is adapted from the Amazon QA dataset [9]. AmazonQA has an answerability classifier predicting whether a question is answerable given the context information. However, we empirically find out that the classifier is of limited precision, i.e. it marks lots of unanswerable questions as answerable. Therefore, we manually inspect all questions and contexts and only include answerable questions.

## B   More Experimental Results and Analyses

### B.1   Model Access

Table 13 shows the model checkpoints and API versions we use for experiments. We set temperature as 0 for all evaluations to try to eliminate the impact of randomness.

## B.2 Hardware Platform

Our experiments are performed on AWS EC2 instances. Two types of instances are used,

- For models with about 70B parameters, we adopt p4d.24xlarge instances with $8\times$ NVIDIA A100 (40GB) GPUs.
- Otherwise, we use g4dn.12xlarge instances with $4\times$ NVIDIA T4 (16GB) GPUs.

We also perform experiments on TACC [46] equipped with $8\times$ NVIDIA 3090 (24GB) GPUs.

We did not closely track the total amount of compute used, but as a reference, a 7B model takes roughly 4 hours to finish inference on Shopping MMLU with the g4dn.12xlarge instance, while a 70B model takes roughly 10 hours to finish inference on a p4d.24xlarge instance.

## B.3 Tasks with Negative Correlations

We list tasks pairs with negative correlations in Table 14. We observe that all tasks pairs with negative score correlations involve one generation task (underlined). Thus, we hypothesize that the negative correlations can be partially attributed to the metrics for generation tasks (i.e. sentence transformer similarity). Indeed, for generation tasks, the reference text may not be perfect, and a generation dissimilar with the reference is not necessarily bad.

We verify the hypothesis with a case study on the task of "Attribute Naming from Description" in Table 15.

1. In the example of "Inside Diameter", it is clear that ChatGPT performs the worst because it did not closely follow the instructions. However, sentence transformer ranks it favorably against all other models, probably due to the common 2-gram 'inside diameter'.

2. In the example of "Power Plug", human evaluators generally prefer 'power plug type' as it resembles the name of an attribute more. However, the reference text is 'power plug', and thus ranks it over 'power plug type', which goes against human preference.

3. In the example of 'Number of Pieces', human evaluators generally prefer 'quantity per unit' as 'unit' is mentioned in the description. However, the reference text (Number of Pieces) does not include the information of 'unit', and thus, the answer 'quantity per unit' is ranked worst among all answers.

We thus believe that the current metric of sentence transformer similarity still fails to accurately reflect human preference on text generation tasks at times, which we leave as future work.

## B.4 More Results on General Domain IFT

We show the impact of general domain IFT on all 4 skills of Shopping MMLU in Figure 10 with similar observations:

- General domain IFT improves the performance of LLaMA2 and LLaMA3 models on 17 out of 20 model-skill pairs, showing its effectiveness in the majority of cases.
- LLaMA3 models generally benefit more from general domain IFT, which should be attributed to the better quality of IFT data.
- Across all 4 skills, we observe that stronger base models generally benefit less from general domain IFT. In addition, among LLaMA2 models, we observe 3 cases where IFT hurts the performances (LLaMA2-70B/Reasoning, LLaMA2-70B/Multi-lingual, LLaMA2-13B/Multi-lingual), showing that the stronger the base model is, the more likely general domain IFT will have a negative impact.

## B.5 More Results on In-context Learning

We select representative tasks to study in-context learning based on the score correlation between a task and the skill it belongs to. The higher the correlation, the more representative the task is of the skill. The selected tasks and their score correlations with their skills are shown in Table 16.

Table 13: Specific model versions used in the experiments.

| Model Type | Name | Platform | Version |
|---|---|---|---|
| Proprietary | ChatGPT | OpenAI | `gpt-3.5-turbo-0125` |
| | Claude-3 Sonnet
Claude-2 | AWS Bedrock | `anthropic.claude-3-sonnet-20240229-v1:0`
`anthropic.claude-v2` |
| Open-source | LLaMA2-(7B/13B/70B)
LLaMA2-(7B/13B/70B)-chat
Vicuna-(7B/13B)
LLaMA3-(8B/70B)
LLaMA3-(8B/70B)-Instruct
QWen1.5-(4B/7B/14B/70B)
Mistral-7B
Mistral-7B-Instruct
Mixtral-8x7B
Zephyr
Phi-2
eCeLLM-(S/M/L) | HuggingFace | `meta-llama/Llama-2-<size>-hf`
`meta-llama/Llama-2-<size>-chat-hf`
`lmsys/vicuna-<size>-v1.5`
`meta-llama/Meta-Llama-3-<size>`
`meta-llama/Meta-Llama-3-<size>-Instruct`
`Qwen/Qwen1.5-<size>`
`mistralai/Mistral-7B-v0.1`
`mistralai/Mistral-7B-Instruct-v0.2`
`mistralai/Mixtral-8x7B-v0.1`
`HuggingFaceH4/zephyr-7b-beta`
`microsoft/phi-2`
`NingLab/eCeLLM-<size>` |

Table 14: Task pairs with negative correlations. Generation tasks are underlined.

| Task 1 | Task 2 | Correlation |
|---|---|---|
| Product Category Explain | Review Rating Prediction | -0.1105 |
| Product Category Naming from Description | Review Rating Prediction | -0.0194 |
| Attribute Naming from Description | Single Conversation Topic Selection | -0.077 |
| Attribute Naming from Description | Query-product Ranking | -0.0005 |
| Attribute Naming from Description | Review Rating Prediction | -0.2453 |
| Attribute Naming from Description | Multi-lingual Query-product Ranking | -0.0469 |
| Aspect-sentiment-based Review Generation | Multi-lingual Query-product Ranking | -0.0279 |

Table 15: Case Study of the task "Attribute Naming from Description". The reference answers are given in brackets and underlined.

| Prompt | Model | Outputs | Score |
|---|---|---|---|
| Instruction: Name the attribute according to the following description with up to 3 words. Input: It provides the inside diameter of this product Output: (Inside Diameter) | Claude-2 | inner diameter | 0.861 |
| | Claude-3 | inner diameter | 0.861 |
| | ChatGPT | inside diameter: 3 inches | 0.889 |
| | GPT-4 | inner diameter | 0.861 |
| Instruction: Use no more than 3 words to name the attribute given the following description. Input: It represents the type of power plug the item supports Output: (Power Plug) | Claude-2 | power plug type | 0.859 |
| | Claude-3 | power plug | 1 |
| | ChatGPT | power plug type | 0.859 |
| | GPT-4 | plug type | 0.671 |
| Instruction: Name the attribute according to the following description with up to 3 words. Input: It represents the total number of pieces included in the unit Output: (Number of Pieces) | Claude-2 | quantity per unit | 0.406 |
| | Claude-3 | item count | 0.556 |
| | ChatGPT | quantity or pieces | 0.745 |
| | GPT-4 | quantity per unit | 0.406 |

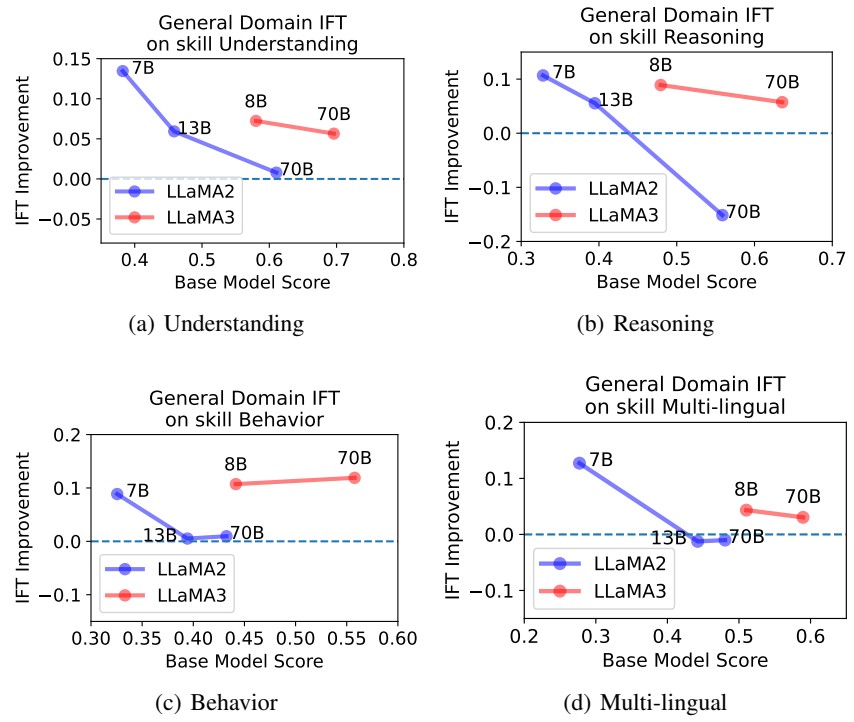

(a) Understanding

(b) Reasoning

(c) Behavior

(d) Multi-lingual

Figure 10: The impact of general domain IFT on all 4 skills of Shopping MMLU.

Table 16: Selected tasks for in-context learning and their correlations with their skills.

| Skill | Task | Score Correlation |
|---|---|---|
| Shopping Concept Understanding | Attribute Value Synonym | 0.8457 |
| | Applicable Product Category Selection Given Attribute | 0.9053 |
| | Aspect-based Sentiment Classification | 0.9010 |
| | Aspect-sentiment-based Review Retrieval | 0.948 |
| | Attribute Value Extraction | 0.9227 |
| | Review Aspect Retrieval | 0.894 |
| | Product Keyphrase Retrieval | 0.8835 |
| Shopping Knowledge Reasoning | Product Numeric Reasoning | 0.8394 |
| | Product Compatibility | 0.8815 |
| | Related Brands Selection | 0.9307 |
| User Behavior Alignment | Query-query Intention Selection | 0.9074 |
| | Session-based Query Recommendation | 0.8622 |
| | Product Co-purchase Retrieval | 0.9395 |
| Multi-lingual Abilities | Multi-lingual Product Keyphrase Selection | 0.8332 |
| | Cross-lingual Entity Alignment | 0.9389 |
| | Multi-lingual Session-based Product Recommendation | 0.7896 |

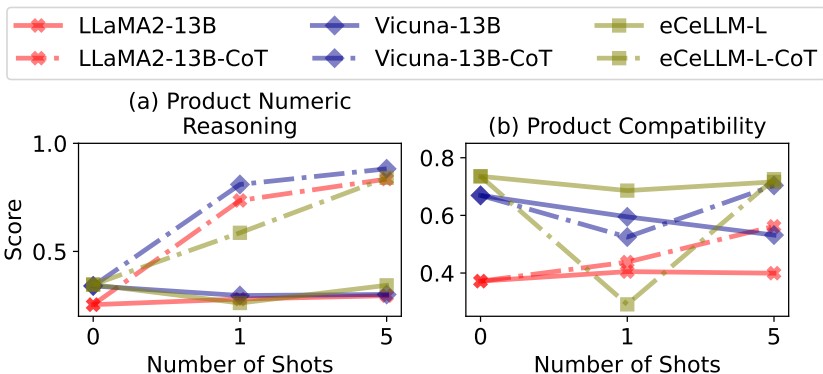

Figure 11: Results of in-context learning (0-, 1-, and 5-shot, with and without CoT) on representative reasoning tasks in Shopping MMLU.

In addition to ordinary few-shot prompting which simply puts question-answer pairs in the prompt, we also explore the effects of few-shot chain-of-thought (CoT) [41] prompting. Specifically, we generate reasoning processes with GPT-4 and manually check their correctness, and put questions, reasoning processes, and final answers in the prompts. We apply CoT prompting on two reasoning tasks, Product Numeric Reasoning and Product Compatibility, and show results in Figure 11. We observe that CoT prompting significantly boosts the performances on numeric reasoning (Product Numeric Reasoning, Figure 11(a)). However, its effects on implicit multi-hop reasoning (Product Compatibility, Figure 11(b)) is mixed, especially between 1-shot and 5-shot learning. Nonetheless, with 5-shot CoT prompting, all models achieve some improvements, showing that CoT prompting is generally helpful in enhancing the reasoning ability of LLMs, while the naive few-shot prompting fails.

Table 17: Comparison between LLMs and task-specific state-of-the-art methods.

| Method | Aspect-based Sentiment Classification | Query-product Relation Selection | Query-product Ranking |
|---|---|---|---|
| Task-specific | **0.8627** | **0.6071** | **0.8846** |
| ChatGPT | 0.8235 | 0.4036 | 0.8374 |
| Claude-3 | 0.7745 | 0.4321 | 0.8491 |
| Claude-2 | 0.8235 | 0.4000 | 0.8147 |

## C Future Work and Limitations

With a comprehensive evaluation of LLMs in online shopping, Shopping MMLU opens up a broad horizon of future work. Specifically, we highlight the room for improvement by showing that existing LLMs are still lagging behind task-specific state-of-the-art methods with three examples. Detailed results are shown in Table 17.

- **Aspect-based Sentiment Classification**, which is a typical task in fine-grained understanding of user reviews. We compare state-of-the-art LLM solutions with the pre-trained model in PyABSA [47][3]. We only compare on reviews with 'positive' and 'negative' sentiments as the other two choices ('mixed', and 'the aspect is not mentioned') are not covered in PyABSA. As shown, the pre-trained model in PyABSA outperforms all proprietary LLMs.

- **Query-product Relation Selection**, which is a typical task in understanding user queries and shopping intentions. We compare LLM solutions with an open-source solution from KDD Cup 2022 [44][4], which ranks 2nd in this task. As shown, the task-specific method

---

[3]https://huggingface.co/yangheng/deberta-v3-large-absa-v1.1
[4]https://gitlab.aicrowd.com/wufanyou/kdd_task_2

outperforms all proprietary LLMs by a significant margin. We note that Shopping MMLU data for this task are sampled from the *test set* of KDD Cup 2022, and thus there is no risk of data leakage.

- **Query-product Ranking**, which is a crucial task in improving the browsing experience of users. We also compare LLM solutions with the solution from KDD Cup 2022 [44], which ranks 6th in this task. Similarly, all proprietary LLMs perform worse than the task-specific method. Similarly, as Shopping MMLU data is sampled from the *test set* of KDD Cup 2022, there is no risk of data leakage.

Therefore, significant efforts are still needed to advance the performance of LLM-based multi-task solutions beyond task-specific ones in online shopping, such as more diverse continue pre-training and IFT datasets with higher quality. Another interesting direction is to build an LLM-based online shopping agent that adaptively routes a question to its corresponding task-specific method.

In addition, as the characteristics of online shopping in Figure 1, i.e. domain-specific concepts, implicit knowledge, human behaviors, and multi-linguality are not unique but apply to a wide range of specific domains (e.g. code [35], education [17], psychology [55], etc.), we believe that Shopping MMLU provides a testbed for future research and development efforts that build domain-specific LLMs in general, such as data mixing strategies, mitigating catastrophic forgetting, knowledge-selective training, retrieval-augmented generation (RAG), etc. We also believe that the insights uncovered in this work effectively lower the technological barrier of developing LLM-based applications, making it more accessible and inclusive to the community.

We finally discuss the limitations of our work. First, we acknowledge that even though we perform manual inspections, label errors may still exist in Shopping MMLU due to subjective human knowledge, preferences, and behaviors. Second, Shopping MMLU primarily focuses on the purpose of evaluation, and thus we do not provide a diverse IFT dataset in online shopping in this work. We identify an equally diverse IFT dataset as Shopping MMLU for future work. Finally, despite our efforts to include as many tasks and skills as possible, our efforts are mostly limited to Amazon data. Therefore, Shopping MMLU, as well as the insights revealed may not accurately reflect online shopping behaviors in other platforms.

