# OpenReview forum: "Shopping MMLU: A Massive Multi-Task Online Shopping Benchmark for Large Language Models"
_NeurIPS.cc/2024/Datasets_and_Benchmarks_Track — NeurIPS 2024 Track Datasets and Benchmarks Poster_

### Official Review · Reviewer_a2Ap · 2024-06-30
**Review for paper 511**

**Rating:** 6
**Confidence:** 3
**Correctness:** yes

**Review:**

The paper proposed ShopBench to  involve a wide range of tasks, entities, and user behaviors, and verified the effectiveness of the data set by extensive experiments.
Pros:
1. The paper addressed the specific challenges of LLM solutions for online shopping, and the proposed data set is designed to handle the specific issues;
2. ShopBench covers a wide range of tasks and skills essential for online shopping, ensuring a thorough evaluation of LLMs in this domain;
3. The benchmark is derived from real-world Amazon data, enhancing the applicability of the evaluation.
4. The evaluation of over 20 LLMs(both proprietary and open-source LLMs) provides valuable insights into the current state and future prospects of LLM-based shopping assistants.

Cons:
1. The uniqueness of the proposed data set is limited, even the proposed data set is designed for the specific challenges of LLM solutions for online shopping, but most of the challenges are covered or partially covered in the previous work;
2. The experiment shows that the proposed data set is a challenging benchmark for LLM evaluation in online shopping solution, but the effectiveness for the fine tuning part for further improvement is limited;

**Strengths:**

As described in the review part

**Additional Feedback:**

see Opportunities For Improvement part.

**Clarity:**

Overall, it is well written but I think the motivation and conclusion part can be articulated more clearly

**Documentation:**

yes

**Limitations:**

see Opportunities For Improvement part.

**Opportunities For Improvement:**

While it is an insightful analysis and experimental results, and the authors mentioned some findings and short comings, but they failed to provide real conclusion and  future solution for the issues. Also for some of the tasks mentioned, such as query-product ranking, LLM may not be the only factor to be considered, how to combine the LLM with current ranking model is also important in real application.

**Relation To Prior Work:**

The paper quotes the previous used publicly available data set and explained the common and different parts of each data set.

**Summary And Contributions:**

The paper presents ShopBench, a comprehensive benchmark designed to evaluate LLMs in the context of online shopping. Traditional models and benchmarks are typically tailored to specific tasks, which limits their ability to capture this complexity. ShopBench addresses these challenges by providing a diverse set of 57 tasks that encompass four key shopping skills: concept understanding, knowledge reasoning, user behavior alignment, and multilinguality. In order to verify the effectiveness of the proposed data set, they conduct extensive experiments with both proprietary and open-source LLMs. The results reveal that the proposed dataset is a challenging benchmark. Furthermore, they also investigate the impact of IFT and in-context learning with the proposed dataset.

---

> ### Author Rebuttal · Authors · 2024-08-15
>
> **Q1: The uniqueness of the proposed data set is limited. Most challenges are covered or partially covered in related works.**
>
> A1: We agree that we curate ShopBench partially from existing datasets (as listed in Appendix B.1.1). However, we would like to highlight that they occupy a small portion of tasks in ShopBench (~15 out of 57), and thus, most data and tasks in ShopBench are newly curated and not covered by existing works. In addition, as shown in Table 1, compared to existing multi-task online shopping benchmarks like EComInstruct and ECInstruct, ShopBench features a wider range of skills and tasks. On the contrary, EComInstruct primarily focuses on shopping concepts, while ECInstruct mainly captures user behaviors. The broader coverage of skills and tasks poses new challenges beyond existing benchmarks, as eCeLLMs achieve state-of-the-art performances on ECInstruct, but not on ShopBench (Section 4.3, Table 2). Therefore, we believe that ShopBench still offers practical values as a comprehensive and holistic benchmark in online shopping.
>
> **Q2: The effectiveness for the fine-tuning part for further improvement is limited. The authors also did not explore combining LLMs with existing ranking models.**
>
> A2: We acknowledge that the primary focus of ShopBench lies in evaluation, instead of training, which is also stated in Appendix D as a major limitation. However, we believe that our evaluations can also shed light on how to improve the performance of LLMs in online shopping. For example, in Section 4.5.1, we show that general knowledge and abilities transfer well to the specific domain of online shopping. In Section 4.5.2, we show that a comprehensive domain-specific fine-tuning dataset is required. Otherwise, improvements can only be expected on tasks and skills covered in the IFT dataset. Combining these two observations, we believe that the best way to comprehensively enhance LLMs in online shopping is to first conduct extensive continue pre-training with both shopping and general data, and then conduct fine-tuning with a dataset covering a sufficiently broad range of tasks. However, constructing such datasets requires significant extra effort and is beyond the scope of this paper.
>
> We show in Appendix D that LLMs alone are not state-of-the-art ranking models. However, we note that as online shopping requires significant general knowledge (Section 4.5.1), LLMs can be potentially combined with existing recommender systems to boost the understanding of contents and products. In fact, many recent works have already made progress in combining LLMs with ID-based ranking/recommendation models (Kim et al. 2024; Liao et al. 2024; Lin et al. 2024), demonstrating the potential. Designing state-of-the-art LLM-based ranking systems is an important future work, but is also beyond the scope of this work.
>
> (Kim et al. 2024) Large Language Models meet Collaborative Filtering: An Efficient All-round LLM-based Recommender System. KDD 2024
>
> (Liao et al. 2024) LLaRA: Large Language-Recommendation Assistant, SIGIR 2024
>
> (Lin et al. 2024) ClickPrompt: CTR Models are Strong Prompt Generators for Adapting Language Models to CTR Prediction. WWW 2024

---

> > ### Comment · Reviewer_a2Ap · 2024-09-01
> >
> > I thank the authors for their detailed response.  I will keep my original score of 6.

---

### Official Review · Reviewer_ePES · 2024-07-20
**A well developed Online Shopping Benchmark**

**Rating:** 7
**Confidence:** 4

**Review:**

This paper overall demonstrates high quality. The paper proposes a novel multi-task online shopping benchmark called ShopBench, which not only provides a brand-new benchmark for evaluating the performance of large language models on complex shopping tasks, but also utilizes real-world data from Amazon, enhancing the practicality and real-world relevance of the research.
In terms of quality, the paper showcases a comprehensive evaluation of existing large language models through in-depth experimental analysis and rigorous metric selection. Regarding clarity, the paper has a well-structured and logical flow, from the problem background to methodology, experimental design, result analysis, and finally the conclusion and future work. The clear categorization of tasks and skills, as well as the detailed experimental setup and results presentation, further enhance the readability and professionalism of the paper. The paper mentions that ShopBench was used for the KDD Cup 2024 competition and promises to open-source ShopBench after the competition.
One limitation of ShopBench is the data selection, as it only includes Amazon data and may not fully reflect the user habits of other e-commerce platforms.

**Strengths:**

1. ShopBench uses real-world Amazon data and has designed 57 different downstream tasks for various use cases, providing a comprehensive and practical framework to evaluate the performance of large language models on online shopping tasks.

2. The paper evaluates recent mainstream large language models on ShopBench and conducts zero-shot experiments, validating the usefulness of the ShopBench dataset.

**Additional Feedback:**

None.

**Clarity:**

The paper is generally clear and easy to read overall. However, in Section 4.5 How to Build LLM-based Shop Assistants, the details on the construction of the Assistants are not explicitly articulated.

**Correctness:**

The experimental design of this paper appears to be reasonably sound. It covers the major LLMs that have emerged in recent years, including models with varying parameter sizes. The experiments utilize appropriate evaluation metrics tailored to the different downstream tasks. Furthermore, the study also includes zero-shot experiments.

**Documentation:**

The paper claims that ShopBench was used for the KDD Cup 2024 competition, and that it will be open-sourced after the competition concludes.

**Ethics:**

None.

**Limitations:**

The paper acknowledges the limitations discussed in Appendix D, including potential biases in the dataset, the inadequacy of the evaluation metrics. Additionally, the dataset may contain label errors due to the subjectivity of users. Since ShopBench is primarily based on Amazon data, it may not be able to fully capture the online shopping behaviors across other e-commerce platforms.

**Opportunities For Improvement:**

The fact that ShopBench is primarily built based on Amazon data may limit the representativeness of its coverage of online shopping behaviors across different regions and e-commerce platforms globally. This could potentially restrict the general applicability of the model evaluation results obtained using ShopBench.

**Relation To Prior Work:**

The paper clearly articulates how it differs from prior work. In the introduction, it mentions that online shopping is a complex multi-task learning problem, and points out that existing models and benchmarks are typically designed for specific tasks, failing to fully capture the full complexity of online shopping. In the related work section, the authors summarize the existing online shopping datasets and note that they usually focus on one or a few closely related tasks, without reflecting the multi-tasking nature of online shopping. Through this comparative analysis, the paper highlights the unique position of ShopBench as a multi-task dataset.

**Summary And Contributions:**

The paper introduces ShopBench, a large-scale multi-task online shopping benchmark designed for large language models (LLMs). ShopBench contains 57 tasks, covering four key shopping skills: conceptual understanding, knowledge reasoning, user behavior alignment, and multilingual capability. The goal is to comprehensively evaluate the ability of LLMs to serve as general-purpose shopping assistants. ShopBench is built using real data from Amazon, providing a standardized testing platform to drive research and applications of LLMs in the online shopping domain.

---

> ### Author Rebuttal · Authors · 2024-08-15
>
> **Q1: The fact that ShopBench uses primarily Amazon data may limit the applicability of the model evaluation results obtained using ShopBench.**
>
> A1: We acknowledge the limitation that ShopBench relies mostly on Amazon data, and thus user behaviors in ShopBench may not accurately reflect those on other shopping platforms (Appendix D). However, the products, attributes, and related knowledge are mostly shared across platforms. Moreover, the multi-lingual skill of ShopBench can also enhance its applicability across different markets. Therefore, we believe ShopBench still has abundant practical value in evaluating the ability of LLMs in online shopping.

---

> > ### Comment · Reviewer_ePES · 2024-08-27
> >
> > Thanks for the author's response. I agree that ShopBench can evaluate the ability of LLMs in the online shopping. However, the data being solely sourced from Amazon remains a limitation of ShopBench.

---

> > > ### Author Response · Authors · 2024-08-27
> > > **Thanks for your reviewing efforts.**
> > >
> > > We thank the reviewer for the reviewing efforts. It is unfortunate that the reviewer lowered the score after our rebuttal. However, we still appreciate the reviewer for pointing this out. We will strengthen the discussion of limitations in the future version of this paper.

---

### Official Review · Reviewer_VJmm · 2024-08-04
**interesting paper, but need to differentiate itself from a pure LLM Eval dataset.**

**Rating:** 8
**Confidence:** 3
**Clarity:** Yes. The paper is generally well writ…

**Review:**

See strengths and weaknesses

**Strengths:**

1. Well written and easy to follow.
2. It comprises a substantial number of tasks (57) from a systematic perspective (4 aspects), utilizing a large-scale validation dataset (20,799 questions).
3. Extensive evaluations are conducted on multiple LLMs of varying scales (ranging from 2.7B to 70B parameters), with or without IFT. These results provide valuable insights for future applications of LLMs in online shopping.

**Additional Feedback:**

N/A

**Correctness:**

Related to weakness 2. While being satisfied with the data collection and quality control, my major concern lies in how these aspects are compiled together from a online shopping point of view, rather than building an evaluation around the LLM. There are many general LLM benchmarks related to long-tail concepts, domain knowledge and reasoning ability, thus I did not expect to encounter "yet another evaluation paper on LLM."

**Documentation:**

Yes. Sufficient detail are provided.

**Ethics:**

No.

**Limitations:**

The authors consider most of major limitations of this work.

**Opportunities For Improvement:**

1. Some results are slightly counterintuitive and require further explanation. For instance, the claim in subsection 4.5.1, "General Knowledge Transfers Well to Online Shopping," slightly conflicts with the conclusion related to ICL. The conclusion regarding ICL indicates that the internally encoded general knowledge is not better triggered by providing demonstrations of domain-related tasks, which is counterintuitive if general knowledge transfers well.
2. How are these 4 aspects and 57 tasks selected, and how do they integrate into the fundamental pipeline of online shopping? These 4 aspects appear to be sampled based on the abilities of LLMs rather than constructed from first principles. While I understand that the primary focus is to evaluate LLM-based online shopping agents, I would prefer to see a more systematic approach to what constitutes a general online shopping system and then identify where LLMs can serve as better alternatives.
3. The observations about "domain-specific models are not always strong" need revision. A fair comparison should be between the domain-specific models and their base counterparts (e.g., eCeLLM-M vs. Mistral-7B, eCeLLM-L vs. LLaMA2-13B). From this perspective, the domain-specific versions outperform the base versions at scales larger than 7B by a significant margin.

**Relation To Prior Work:**

Yes. If I understand correctly, the most related work are multi-task online shopping dataset to validate LLM-based shop assistant from all aspects, such as EComInstruct and ECInstruct mentioned in related work. The authors have discussed the difference between this work and all these previous works.

**Summary And Contributions:**

This paper introduces a benchmark dataset designed to evaluate the performance of large language model (LLM)-based shop assistants. The dataset is divided into four categories, encompassing 47 distinct tasks. Various LLMs, both open-source and proprietary, are rigorously tested against these benchmarks. The study distills valuable insights from the extensive evaluation, offering a comprehensive understanding of the capabilities and limitations of these models in the context of shop assistant applications.

---

> ### Author Rebuttal · Authors · 2024-08-15
>
> **Q1: The observation that ICL does not work well is counter-intuitive.**
>
> A1: We would like to further explain the observation from two aspects.
>
> |Model| Avg. 1-shot improvement | Avg. 5-shot improvement|
> |-------|---------| -----|
> |ChatGPT| 4.08\% |3.16\%|
> |QWen1.5-72B| 1.45\% | 3.58\%|
> |LLaMA2-13B| 1.07\% | 1.58\%|
> |Vicuna-13B|0.41\% | -0.18\%|
> |eCeLLM-L| -5.36\% | 3.92\%|
>
> We first show the average improvements achieved by in-context learning in the table above. As shown, ICL boosts the performances of almost all models (except for eCeLLM) on ShopBench on average, showing that ICL is still generally effective in triggering the encoded general knowledge in the models. However, the effectiveness of ICL is known to vary significantly across different tasks, few-shot examples, etc. Thus, ICL may not be effective on every single task or skill, which is what we show in Section 4.5.3. The inability of eCeLLM-L to learn from ICL may be explained by the observations by (Shin et al. 2022) that training on multiple domains yields better ICL abilities than training on a single domain, which is what eCeLLM-L does (fine-tuning with only e-Commerce data).
>
> Second, the effectiveness of ICL depends strongly on the prompts. In Section 4.5.3 we follow the standard few-shot prompts (i.e. directly appending answers after the question), which was ineffective for reasoning tasks. However, as we show in Appendix C5, after switching to few-shot chain-of-thoughts prompting, we observe significantly more effective ICL for LLaMA2-13B, Vicuna-13B, and eCeLLM-L on reasoning tasks, showing that the general knowledge in the models is still effective, but may need more elaborate prompts to activate. Designing elaborate prompts, however, is beyond the scope of this paper.
>
> **Q2: How are the skills and tasks selected, and what role do they serve in the general pipeline of an online shopping system.**
>
> A2: The skills in ShopBench are selected in two directions. The first direction, which is presented in the paper, originates from the challenges faced by LLMs in online shopping and eventually leads to the four skills in our paper. The second direction originates from the requirements of online shopping, which boils down to two basic requirements: understanding products and understanding users. Understanding users corresponds to the skill of ‘user behavior alignment’ exactly, while understanding products corresponds to ‘shopping concept understanding’ and ‘shopping knowledge reasoning’, where we separate ‘reasoning’ from ‘understanding’ due to its need for complex knowledge and multi-step derivations. Therefore, both directions lead to very similar skill requirements, which are what we show in our paper.
>
> The tasks in each skill are selected from real-world problems in Amazon. For example, tasks related to product concepts (e.g. product category, attributes, and attribute values) are sampled from the Amazon product graph (https://www.amazon.science/blog/building-product-graphs-automatically), which is used in practice for data imputation, mislabeling detection, and relation discovery; review-related tasks (e.g. sentiment classification, keyphrase extraction) are sampled from opinion mining tasks used in Amazon production; query-related tasks (e.g. related keyword, query re-writing) are real-world production tasks that improve customer search and shopping experiences, etc. With these tasks, ShopBench aims to provide an extensive evaluation benchmark that reflects the needs of a real-world online shopping system.
>
> Our evaluation indicates that different tasks in online shopping share a large amount of knowledge in common (Section 4.4), and that online shopping requires strong world knowledge and general abilities (Section 4.5.1). Therefore, although LLMs cannot achieve state-of-the-art performances in an end-to-end manner so far (Appendix D, Table 17), their rich general abilities enable them to be versatile feature extractors for various online shopping tasks, as shown by various recent works that combine ID-based recommender systems with LLMs (Kim et al. 2024; Liao et al. 2024; Lin et al. 2024).
>
> **Q3: The observations about "domain-specific models are not always strong" need revision.**
>
> A3: We apologize for the confusion. We actually made two observations related to domain-specific models throughout the paper.
>
> - In Section 4.3 (Overall Performance), we compare domain-specific eCeLLM models with the SOTA proprietary and open-source models, where we find eCeLLM models not as strong as these SOTA counterparts. We make such a comparison because the eCeLLM paper [32] claims that eCeLLM models outperform GPT-4 on their benchmarks, and thus we follow the same baselines (i.e. SOTA LLMs like GPT-4 and LLaMA3).
> - In Section 4.5.2, we compare eCeLLMs with their base models (Figure 6), where eCeLLMs perform better than their general domain base models, but mostly on seen skills and tasks. These results echo your observation that domain-specific fine-tuning is effective (although not SOTA) in improving performances in the same domain.
>
> To conclude, our evaluations show that domain-specific fine-tuning and the resultant eCeLLMs are stronger than their general domain base models, but not as strong as SOTA (as claimed by their original paper). We will revise related observations to clarify the confusion.
>
>
>
> (Shin et al. 2022) On the Effect of Pretraining Corpora on In-context Learning by a Large-scale Language Model. NAACL 2022
>
> (Kim et al. 2024) Large Language Models meet Collaborative Filtering: An Efficient All-round LLM-based Recommender System. KDD 2024
>
> (Liao et al. 2024) LLaRA: Large Language-Recommendation Assistant, SIGIR 2024
>
> (Lin et al. 2024) ClickPrompt: CTR Models are Strong Prompt Generators for Adapting Language Models to CTR Prediction. WWW 2024

---

> > ### Comment · Reviewer_VJmm · 2024-08-16
> > **Thanks for your clarification**
> >
> > The response is comprehensive and effectively addresses most of my concerns. Please ensure that the clarification of Q3 is included in the final version of the paper.  In conclusion, The benchmark is well-developed, and the experiments offer valuable insights for further developing the use of LLMs as Online Shopping Assistants. I appreciate your effort and am pleased to raise my overall rating from 6 to 8.

---

> > > ### Author Response · Authors · 2024-08-17
> > > **Thanks for your efforts and recognition.**
> > >
> > > We appreciate your efforts in reviewing our paper and providing us with constructive feedback. We also appreciate your prompt reply and recognition of our work.
> > >
> > > The clarification will be included in the revised version of our paper.
> > >
> > > Best,
> > >
> > > Paper 511 Authors

---

### Official Review · Reviewer_qCzS · 2024-08-04
**Novel, massive-scale shopping task benchmark that extensively covers a number of tasks**

**Rating:** 6
**Confidence:** 4
**Correctness:** Yes and yes.

**Review:**

I believe ShopBench is a strong contribution as a dataset + benchmark in the online shopping task space, which has proliferated as a popular web task for evaluating language models in recent years. From my recollection, I believe the scale of the dataset + taxonomy/organization can make usage of this dataset very compelling, and can hopefully support long term investments by the community in this work due to how multi-dimensional and deep it is.

My main feedback for this work would be twofold. First, I think the related work, while well carried out in its writing and comparison to identified works, misses out on the most important, existing shopping task benchmarks. Without these comparisons, the arguments for ShopBench’s novelty and contribution feel not-as-strong. I put more details in the “Relation to Prior Work” section of this paper.

Second, the takeaways in Section 4 are sound and make sense, but do not provide any particularly interesting insight that was not already reflected by prior works. It is also a bit concerning that proprietary models can already achieve 80+% on ShopBench tasks. While such a benchmark can be useful for closing the gap between open-source and proprietary models, it may not be as compelling for challenging the cutting edge of existing models.

NOTE: This is unrelated to the paper. I did want to point out that I was able to see the authors’ names + affiliations when performing my review, due to the PDF that was uploaded. I’m not sure whether the review procedure is different for NeurIPS D&B vs. NeurIPS main track, but I did want to be upfront that the authors were visible to me.

**Strengths:**

* 57 tasks and 20799 questions is a broad expanse of tasks that eclipses the scale of prior works (e.g. WebShop, WebArena)
* Many of the tasks capture dimensions of online shopping that were previously not reflected by existing benchmarks. The general completeness of the benchmark’s tasks is quite admirable.
* The curation process and the quality of the questions seem to be quite good, based on the Appendix examples. ShopBench does a great job unifying a lot of separate inquiries in research on different facets of the online shopping task.

**Additional Feedback:**

None

**Clarity:**

The paper is written in a logically sound order, following the general order of introduction, related work, methodology, experiments (broken down into setup discussions, followed up by performance + question-specific analyses), and conclusion. I was able to follow along fairly easily. The way the takeaways were organized in Section 4.5 are fairly well organized.

**Documentation:**

The authors are currently hosting a competition with this dataset on KDD Cup. They mention that data will be released post competition. At this time, from the provided code, there is sufficient code to reflect how experiments were carried out. However, the dataset is not accessible beyond the competition. However, if the authors carry out their release plan as discussed and promised in the paper (Appendix A), there should be no problem with its usability, maintenance, and availability.

**Ethics:**

None stood out to me. Sections 3.1 and 3.3 discuss the collection process at a high level. From the supplementary material + code, I did see that there was experiment code provided, but I don’t believe I found any data cleaning code. It would be great if, in addition to the writing, the authors provided some examples of the scripts used for data collection + cleaning to get a more grounded sense of how data was collected safely + anonymity was preserved. Aside from this ask, I generally didn’t find any ethical concerns with the paper.

**Limitations:**

The authors cover this in Appendix D. I believe it does a good job addressing limitations and clarifying what ShopBench does and does not provide.

**Opportunities For Improvement:**

As mentioned in the overall review, the related work coverage and experimental takeaways left more to be desired.
* More comprehensive related work, in terms of papers covered. I realize that ShopBench is more seq2seq oriented, while WebShop, WebArena are more agent-ic. However, I still feel that capturing the full spectrum of such related work is important to justifying where ShopBench’s impact lies.
* The experimental results are good. However, it was a bit hard to determine why ShopBench is more challenging or involved than other benchmarks, or how the insights that ShopBench reveals about model performance are unique to this benchmark. More demonstration of how ShopBench is an effective measure of model performance compared to other web shopping benchmarks would be helpful.
* I understand that the authors’ focus for this work is more on seq2seq tasks within the shopping world. Is there potentially an opportunity to string together these tasks and datasets under the umbrella of more complicated multi-step tasks that are collectively more challenging for a model to perform in a seq2seq/agent-ic setting? This seems to be the direction that many web shopping tasks are trending towards.

**Relation To Prior Work:**

There is a fair amount of related work that was not referenced:
* WebShop: Towards Scalable Real-World Web Interaction with Grounded Language Agents
* WebArena: A Realistic Web Environment for Building Autonomous Agents
* VisualWebArena: Evaluating Multimodal Agents on Realistic Visual Web Tasks
There are a couple more - I would encourage the authors to look at the cited by’s for these papers to discover if there are additional relevant works.

There are good efforts made to compare ShopBench to the existing datasets. I would recommend the authors extend their discussion to some of the more seminal existing works in this domain. However, as a result of this, I would also argue that some of the contributions the authors claim are not that novel. For instance, in the latter half of the introduction, the authors characterize the dimensions of the online shopping task and compare against existing datasets along these dimensions in Table 1. An updated table with the above works would be helpful.

**Summary And Contributions:**

This paper introduces ShopBench, a new dataset and benchmark that introduces a lot of tasks (57) and task instances (20799) that provides strong coverage in representing the general workflow and intricacies of real world online shopping experiences by humans. The contributions I gleaned from the paper are as follows:
* Performs data collection across a broad range of public + private Amazon data, which includes products, reviews, user interactions
* Provides a higher level taxonomy of 4 shaping skills for categorizing the 57 tasks, including concept/reasoning/behavior/multilingual. The task types refer to the different possible input/output formats, including multiple choice, retrieval, ranking, NER, generation.
* Runs extensive experiments on a large number of proprietary + open-source models, presenting comprehensive results on the performance of different models on these tasks along with insight on the multi-task-ness of ShopBench and the effects of instruction fine-tuning on more specific tasks.

---

> ### Author Rebuttal · Authors · 2024-08-15
>
> **Q1: Comparison with related works on web agents.**
>
> A1: We thank the reviewer for bringing this up. Among the related works mentioned by the reviewer, VisualWebArena requires multi-modal understanding and falls out of the scope of our paper (language models). We discuss the relation between WebShop, WebArena and ShopBench as follows.
>
> **Common:** Both WebShop and WebArena require agents to follow instructions and perform tasks on e-commerce websites, which involves understanding online shopping concepts, e.g. attribute values, product types (I want a small portable folding desk that is fully assembled and has a khaki wood finish.), reviews, product descriptions (Update the product description of Bella Tank to highlight the positive user reviews by quoting them), etc. In addition, reasoning skills like numeric reasoning (The price should be lower than $140, Add the product with the lowest unit price to the cart) are also involved. Finally, WebShop requires agents to perform searches, which resembles the task “Query re-writing” in ShopBench.
>
> **Differences:** We highlight the differences between ShopBench and the related work from two aspects.
> - **Overall Goal.** The overall goal of ShopBench is to evaluate a model’s ability to understand concepts, knowledge, and behaviors in online shopping. On the contrary, agent benchmarks like WebShop and WebArena go one step further to test whether the model can not only understand shopping concepts and knowledge, but also follow instructions and make actions to perform real-world tasks. Therefore, though ShopBench and agent benchmarks (WebShop, WebArena) share similar requirements in the underlying model abilities, their main focuses are different, with ShopBench being more fundamental and agent benchmarks being more complex and interactive.
> - **Coverage of Skills.** Despite requiring some common skills (e.g. understanding shopping concepts, numeric reasoning, etc.), ShopBench captures a broader range of skills than WebShop and WebArena. First, neither WebShop nor WebArena covers multi-lingual abilities — a skill crucial to multi-market platforms like Amazon. Second, neither WebShop nor WebArena covers multi-hop reasoning like product compatibility and complementarity, brand similarity, etc. Finally, agents in both WebShop and WebArena operate as customers, instead of the platform. Therefore, neither contains tasks to understand and model user behaviors (e.g. recommending related keywords and co-purchase products, session-based recommendations, etc. ).
>
> We hope that the discussion enhances the positioning of ShopBench against related work. We will add the discussion to our paper in the revision.
>
> **Q2: The benchmark does not seem more challenging than existing ones.**
>
> A2: We admit that SOTA proprietary and very large open-source models (e.g. QWen-72B, LLaMA3-70B) already perform well on ShopBench, which boils down to our design considerations. Because online shopping applications require fast model responses, there is a trade-off between the inference cost and the model capabilities, and it is often not practical to deploy excessively large models (e.g. 70B). Therefore, we designed ShopBench so that it is neither too hard nor too easy for medium-sized models (e.g. \~14B). Indeed, in Table 2, \~14B models achieve a reasonable performance (\~0.6), but lag behind larger (\~70B) and proprietary ones.
>
> We also believe that ShopBench is still challenging and brings additional insights, which we explain as follows.
>
> The most related work to ShopBench is eCeLLM [32], who curated ECInstruct with both training and evaluation datasets and trained eCeLLM models by fine-tuning general LLMs with ECInstruct. By training and testing on the same set of tasks, eCeLLMs surpass GPT-4 and achieve state-of-the-art accuracies. However, when tested on a different (and much larger) set of online shopping tasks in ShopBench, eCeLLMs perform much worse (shown in Table 2, discussed in Line 196-198) and cannot beat SOTA general-purpose LLMs with similar sizes. We thus consider ShopBench more challenging than ECInstruct, probably due to its much wider range of skills and tasks (Table 1).
>
> We also believe that our experiments did provide unique insights into adapting LLMs to online shopping, such as the transferability of general abilities to online shopping (Section 4.5.1), the effects of general and domain-specific instruction tuning (Section 4.5.2), and the effects of in-context learning (Section 4.5.3).
>
> **Q3: Opportunity to string together ShopBench to form complicated multi-step tasks.**
>
> A3: We designed ShopBench to contain atomic tasks as possible to enable fine-grained skill analysis. However, the need for more complex multi-step tasks is relevant as they better resemble real-world user questions, and such questions can be constructed by bringing tasks in ShopBench together. For example, a user question of “Please recommend shoes that are similar to Asics Gel Nimbus, suitable for men over 90kg, and are rated durable and lightweight by customers” involves the following tasks in ShopBench:
>
> - Named Entity Recognition (extracting the product “Asics Gel Nimbus’)
> - Related Brands (finding brands similar to Asics),
> - Attribute Value Extraction (filter products with attribute “heavyweight runners”)
> - Aspect-based sentiment analysis (summarize reviews on ‘durability’ and ‘weight’)
> - Related Keywords search (Searching for related keywords like ‘running shoes saucony’).
> - Query-product ranking (Re-rank candidates according to the requirements).
>
> We consider a complex, multi-step online shopping benchmark as an important future work, probably by expanding ShopBench with techniques like Evol-Instruct. We also find it promising to build such a benchmark with interactive environments similar to WebShop, but from the perspective of a shop assistant, or a recommendation agent.

---

> ### Author Rebuttal · Authors · 2024-08-15
>
> **Q4: The authors' names and affiliations are visible**
>
> A4: According to the call for papers of NeurIPS Datasets & Benchmarks Track, this track indeed allows single-blind review where authors are visible.
>
> **Q5: Examples of data collection and cleaning scripts**
>
> A5: We apologize for not being able to provide such scripts as they may reveal the schema/data fields of the internal data from Amazon. However, the data cleaning was indeed performed according to what is said in the paper.

---

> > ### Comment · Reviewer_qCzS · 2024-08-23
> > **Thanks for your response**
> >
> > Thanks to the authors for the detailed responses. The response helped with getting a deeper understanding of the authors' perspective on the difficult of the task and how good it may help with driving future model development. It is a bit unfortunate that the full data collection scripts cannot be shared, but it is completely understandable. I will keep my original score of 6.

---

> > > ### Author Response · Authors · 2024-08-27
> > > **Thanks for your reviewing efforts.**
> > >
> > > We thank the reviewer for the reviewing efforts. Although you maintain your score, we believe that your reviews help make this paper better and more complete. We will include the discussions in the future version of this paper.

---

### Author Rebuttal · Authors · 2024-08-15

We appreciate all reviewers for their efforts devoted to reviewing our paper and for the insightful feedback. We are glad that the reviewers find our work

- Captures unique and specific dimensions of online shopping (qCz5, a2Ap)
- Covers a diverse range of skills, tasks, and questions (qCz5, Vjmm, ePES, a2Ap).
- Is curated with real-world Amazon data and is of high quality and practicality (qCz5, ePES, a2Ap).
- Performed extensive experiments and uncovered valuable insights (Vjmm, ePES, a2Ap).
- Is generally well-written and easy to follow (qCzS, Vjmm, ePES, a2Ap).

We are also glad to share the full data as well as related codes for this paper at https://anonymous.4open.science/r/ShopBench-783E/. Some necessary procedures with the Amazon leadership are still underway, and we will publicly release the full data once the procedures are sorted out.

Below your reviews, we provide point-to-point responses to the questions and weaknesses raised by individual reviews. We hope that our response can clear up your questions and concerns. If you come up with additional questions during the discussion, please feel free to raise them, and we will be glad to provide further responses.

Please kindly consider raising your ratings if you feel that your questions are properly addressed.


Best,

Paper 511 Authors.

---

### Decision · Program_Chairs · 2024-09-26

**Decision:**

Accept (Poster)

**Comment:**

This paper receives positive ratings from all reviewers. The dataset, highlighted by the reviewers, is a significant contribution to the research community due to its size, variety of tasks, and comprehensive coverage of user behaviors in the online shopping domain, as represented by Amazon. Some of the weaknesses mentioned in the reviews can be addressed in the next version by incorporating the author’s rebuttal. Additionally, there is potential for this dataset to be used in research beyond just evaluating large language models (LLMs).